# Redox signaling-driven modulation of microbial biosynthesis and biocatalysis

Na Chen[1,4], Na Du[1,4], Ruichen Shen[2], Tianpei He[1], Jing Xi[1], Jie Tan [2], Guangkai Bian[3], Yanbing Yang[1], Tiangang Liu [1], Weihong Tan [2], Lilei Yu[1] & Quan Yuan [1,2]

Microbial communication can drive coordinated functions through sensing, analyzing and processing signal information, playing critical roles in biomanufacturing and life evolution. However, it is still a great challenge to develop effective methods to construct a microbial communication system with coordinated behaviors. Here, we report an electron transfer triggered redox communication network consisting of three building blocks including signal router, optical verifier and bio-actuator for microbial metabolism regulation and coordination. In the redox communication network, the $Fe^{3+}/Fe^{2+}$ redox signal can be dynamically and reversibly transduced, channeling electrons directly and specifically into bio-actuator cells through iron oxidation pathway. The redox communication network drives gene expression of electron transfer proteins and simultaneously facilitates the critical reducing power regeneration in the bio-actuator, thus enabling regulation of microbial metabolism. In this way, the redox communication system efficiently promotes the biomanufacturing yield and $CO_2$ fixation rate of bio-actuator. Furthermore, the results demonstrate that this redox communication strategy is applicable both in co-culture and microbial consortia. The proposed electron transfer triggered redox communication strategy in this work could provide an approach for reducing power regeneration and metabolic optimization and could offer insights into improving biomanufacturing efficiency.

Communication, the sending and receiving of information, is ubiquitous in natural biological system[1,2]. The critical biological information is typically transmitted as diffusion gradients of ions and small molecules during the communication process and plays vital roles in dynamically driving coordinating activities[1,3]. Microbial communication, prevalent among biological systems, has raised great attention owing to its potential in dynamically modifying microbial activities for a purpose in various areas such as biomanufacturing[4–6]. Specifically, microbial consortium can dynamically adjust the gene expression upon dynamically

sensing and dealing with the external signal information through the cell-to-cell communication, thus globally redistributing the cellular resources and coordinate the microbial metabolism processes[7–9]. Consequently, it is of vital significance to have a deep understanding of microbial communication network, thus guiding the microbial metabolism coordination and optimization, offering a promising way for driving microbial industries and better understand living systems.

Currently, several biological communication strategies have emerged including quorum sensing, ion channel, redox-based

[1]Renmin Hospital of Wuhan University, College of Chemistry and Molecular Sciences, Institute of Molecular Medicine, School of Microelectronics, School of Pharmaceutical Sciences, Wuhan University, 430072 Wuhan, P. R. China. [2]Molecular Science and Biomedicine Laboratory (MBL), State Key Laboratory of Chemo/Biosensing and Chemometrics College of Chemistry and Chemical Engineering, Hunan University, 410082 Changsha, P. R. China. [3]Center for Materials Synthetic Biology, Shenzhen Institute of Synthetic Biology, Chinese Academy of Sciences, 518055 Shenzhen, P. R. China. [4]These authors contributed equally: Na Chen, Na Du. e-mail: lileiyu@whu.edu.cn; yuanquan@whu.edu.cn

molecular communication and etc[1,10]. Among the different communication modes, redox-based molecular communication has attracted increasing attention owing to its distinct modality for biological communication[11]. To be specific, molecular redox state can be switched dynamically and reversibly through electron transfer in the redox-based communication process, consequently providing the driving force for dynamically regulation of microbial metabolism[12]. In addition, the activity of signaling enzymes with redox-active functional groups can also be dynamically altered owing to the varied redox potential that is associated with the ratio of oxidized/reduced redox species[1,13,14]. Analogous to the communication network in electronic information technologies, redox-based molecular communication system also has the ability to sense, analyze and process signal information. Hence, constructing microbial redox-based communication network provides an expedient tool for intelligent self-adaptive regulation, enabling the dynamic resource reallocation and high flux in microbial biomanufacturing practice of value-added chemicals[1,15,16].

The $Fe^{3+}/Fe^{2+}$ redox couple, which widely exists in nature, is critical in maintaining metabolic processes[17,18]. Here, we construct a microbial communication network model consisted of three building blocks including signal router, optical verifier and bio-actuator based on microbial electron transfer triggered $Fe^{3+}/Fe^{2+}$ redox reaction (Fig. 1). With the ability to sense, analyze and process the dynamic and reversible Fe redox species, the microbial communication network system is capable of coordinating the metabolism behavior of bio-actuator, leading to an increased pathway flux and improved biomanufacturing efficiency. Specifically, $Fe^{3+}$ redox signal can be sensed and transduced by *Shewanella putrefaciens* CN32 (*S. putrefaciens*) signal router to $Fe^{2+}$ with the metal-reducing (Mtr) pathway[19]. *Rhodopseudomonas palustris* TIE-1 (*R. palustris*), the bio-actuator that is capable to receive and utilize the electrons of the transduced $Fe^{2+}$, has dual functions in $Fe^{2+}$ output signal retranslating with the iron oxidation pathway and lycopene manufacturing with the mevalonate (MVA) and methylerythritol 4-phosphate (MEP) pathway[20,21]. In addition, optical verifier with high spatial and temporal resolution is introduced into the communication model, providing real-time and dynamic monitoring for the fluctuant $Fe^{3+}/Fe^{2+}$ redox signal[22]. We then capture the concept of biological local area network (LAN) which consists of "router" "actuator" and "verifier" as demonstrated in reference[1], and apply the same methodology to a Fe signal based system that has these benefits. In this work, we find that electron transfer related genes are highly expressed in the biological LAN with redox communication. Actually, electrons from $Fe^{2+}$ redox signal can be transferred into *R. palustris* bio-actuator cells with iron oxidation pathway through the electron transfer related proteins (Supplementary Fig. 4)[19], channeling reducing power directionally into *R. palustris* bio-actuator, thus boosting the lycopene biomanufacturing and $CO_2$ fixation efficiency. Further, we demonstrate that biological LAN with redox communication can both coordinate and regulate metabolism behaviors in microbial co-culture (*S. putrefaciens-R. palustris*) and microbial consortia (*S. putrefaciens-Geobacter soli-R.*

*palustris*) system. Compared with the previous work on redox-based communication[1,13], the electron transfer triggered redox communication strategy proposed in this study was independent of external electronics input and the intrinsic electron transfer triggered redox cycle could function as batteries that support microbial metabolism[20]. The redox-based communication strategy in this work may pave ways for programming biological systems with dynamic functions or behaviors and open doors for efficient biomanufacturing.

## Results

### Biological local area network based on redox communication

Actually, the established redox-based communication network above can serve as a compact biological local area network (LAN). Parallel to the electronic LAN, the redox-based communication network has intrinsic advantages including signal propagation, division of labor across populations and spatial organization[1]. Three components of "router" "actuator" and "verifier" in the biological LAN are responsible for the signal interception, relay and interpretation respectively. Specifically, signal router is hardwired to the active redox species via "electron transfer wire", enabling the interception of $Fe^{3+}$ redox signal and transduction into $Fe^{2+}$. Subsequently, $Fe^{2+}$ redox signal can be further transmitted toward the remaining bio-actuator and verifier, thus realizing the signal relay and interpretation independently.

$Fe^{3+}/Fe^{2+}$ couple, as one of the most common redox species in nature, plays crucial roles in geochemical cycle and life evolution through electron transfer triggered redox state switching[18,19]. *S. putrefaciens* is one of the earliest metal-reducing bacteria and is competent to transduce $Fe^{3+}$ redox species into $Fe^{2+}$ signal (Supplementary Fig. 1)[19]. The Mtr pathway in *S. putrefaciens* signal router can be termed as the molecular "central processing unit" (CPU) that is in charge of the redox signal operation and control[17]. By instinct, *S. putrefaciens* signal router transfers electrons to external $Fe^{3+}$ species through the molecular "CPU" consisting of electron transfer proteins such as MtrC and CymA, accomplishing the redox signal transduction[22] (Fig. 2a). To be specific, Mtrc, OmcA and OmcB locate on the bacteria surface and could transfer electrons directly[22], and CymA is a key membrane-anchor protein that could transfer electrons from cytoplasm to periplasm in *S. putrefaciens*[22]. When $Fe^{3+}$ incubated with the *S. putrefaciens* signal router, $Fe^{2+}$ concentration increased over time (Fig. 2b), suggesting the successful redox signal transduction process. Further, mutants with MtrC-encoding gene deleted (ΔMtrC) and CymA-encoding gene deleted (ΔCymA) were constructed to investigate their roles in redox signal transduction process. As shown in Fig. 2c, it can be observed that $Fe^{2+}$ signal transduction efficiency decreases about 1.61 and 1.69 times in ΔMtrC and ΔCymA mutant systems respectively compared with wild-type strain. While, after the addition of electron shuttle humic acid (HA) in *S. putrefaciens* system[23], $Fe^{2+}$ signal transduction efficiency increased 1.23 times (Supplementary Fig. 2). The above results indicate that the redox signal transduction of *S. putrefaciens* signal router is closely related with electron transfer processes.

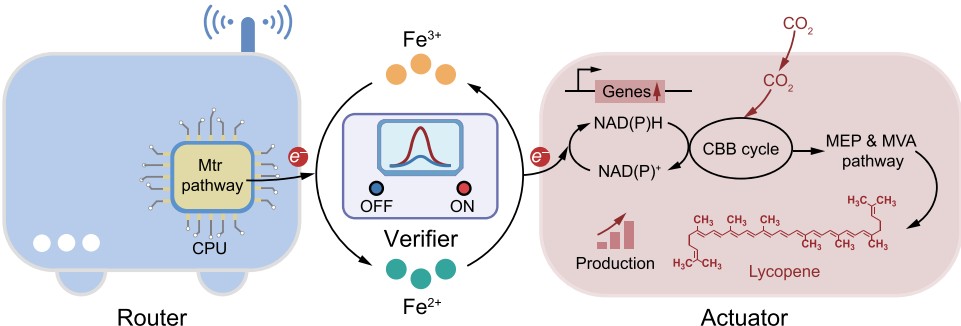

**Fig. 1 | Biological local area network (LAN) with dynamic and reversible redox signal transduction through electron transfer.** The biological LAN consists of three building blocks including signal router, optical verifier and bio-actuator based on microbial electron transfer triggered $Fe^{3+}/Fe^{2+}$ redox reaction.

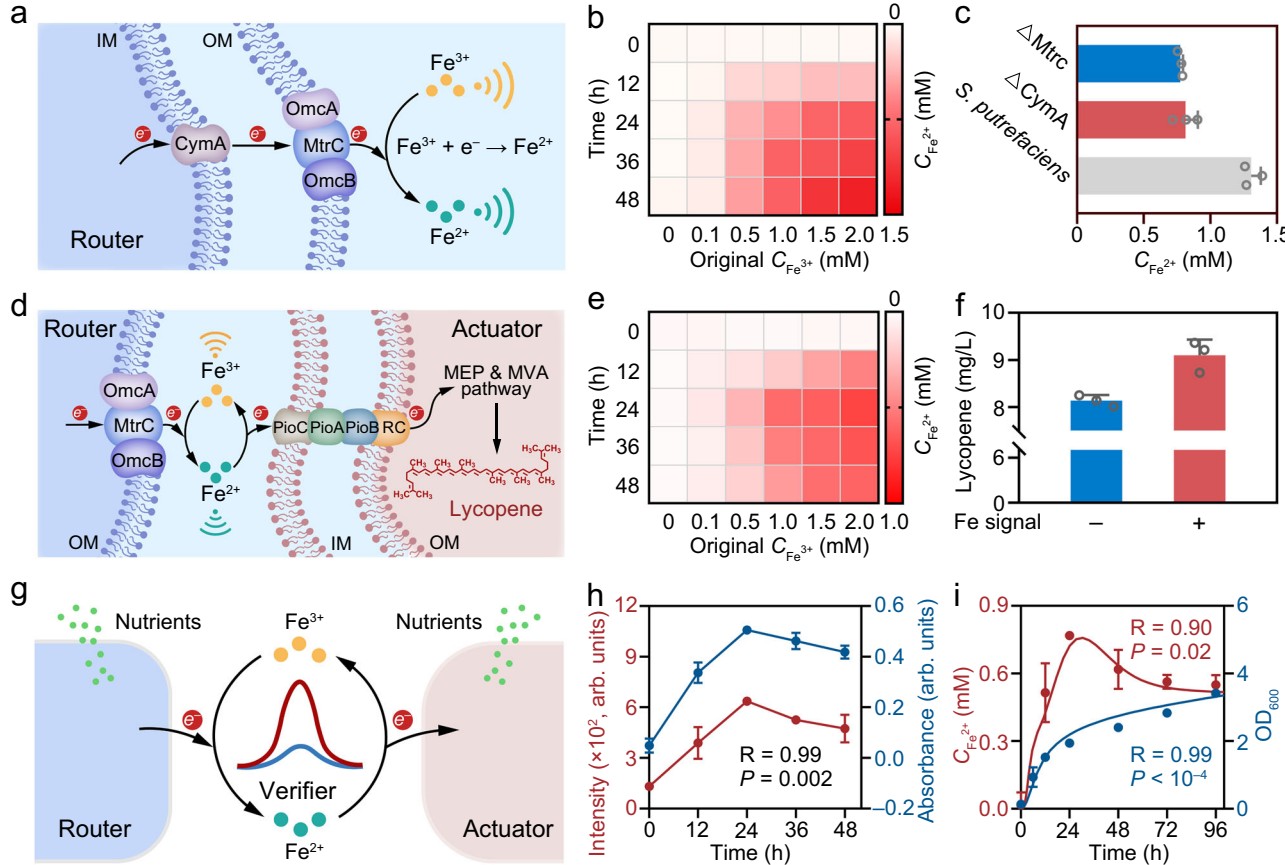

**Fig. 2 | Biological LAN with signal router, optical verifier and bio-actuator.**
**a** Schematic illustration of Fe signal transduction with Mtr pathway in *S. putrefaciens*. **b** $Fe^{2+}$ concentration in bare *S. putrefaciens* system at different incubation time with different initial $Fe^{3+}$ addition. **c** $Fe^{2+}$ concentration in $\Delta MtrC$, $\Delta CymA$ mutants and *S. putrefaciens* system cultured at 48 h. Data presented as mean values ± SD., $n = 3$. **d** Schematic illustration of *R. palustris* with dual actuating functions including signal reconversion and chemical biosynthesis. **e** $Fe^{2+}$ concentration in *S. putrefaciens*-*R. palustris* co-culture system at different incubation time with different initial $Fe^{3+}$ addition. **f** Lycopene biosynthesis yield of *R. palustris* incubated at 96 h with or without Fe redox communication represented by "+" and "−" respectively. Data presented as mean values ± SD., $n = 3$. **g** Schematic illustration of the biological LAN. **h** The recorded persistent luminescence intensity of ZGO:Mn verifier and the absorbance measured with o-Phenanthroline in the biological LAN. $P = 0.002$, $P$ values were determined by a two-tailed test. Data presented as mean values ± SD., $n = 3$. **i** Time course of model-predicted $Fe^{2+}$ concentration and $OD_{600}$ in the biological LAN. Experimental data are indicated by data points and fitted models of $Fe^{2+}$ concentration (red) and $OD_{600}$ (blue) are shown as solid lines. $P = 0.02$, $P$ values were determined by a two-tailed test. Data presented as mean values ± SD., $n = 3$. Source data are provided as a Source Data file.

Next, the transduced $Fe^{2+}$ signal can be further received by bio-actuator and verifier in the biological LAN, realizing the signal relay and interpretation. To be specific, electrons carried by $Fe^{2+}$ signal can be channeled through PioABC complex that is directly involved in taking up extracellular electrons for photoferrotrophic and photoautorophic growth of *R. palustris* bio-actuator (Supplementary Fig. 3)[19]. Meanwhile, when *R. palustris* ingests the electrons, $Fe^{2+}$ signal is reconverted back to $Fe^{3+}$. In addition, *R. palustris*, reported as a microbial host for terpenoid production through the MVA and MEP pathways, possesses potential for high value-added chemical production (Supplementary Fig. 4)[21,24]. Overall, *R. palustris* in the constructed compact biological LAN possess dual actuating functions including signal reconversion and chemical biosynthesis (Fig. 2d). To investigate the functions of the *R. palustris* bio-actuator, we measured the $Fe^{2+}$ concentrations and the biosynthetic yields throughout the incubation period. As shown in Fig. 2e, $Fe^{2+}$ concentration first increases and then decreases over incubation time, mainly owing to the $Fe^{2+}$ signal reconversion process activated by *R. palustris* bio-actuator (Supplementary Fig. 5). Moreover, the lycopene biosynthetic yield of *R. palustris* bio-actuator with Fe redox communication is about 1.12 times higher than the bare co-culture system without Fe redox communication, suggesting that the electron transfer triggered redox communication enhances the actuating function of biosynthesis (Fig. 2f, Supplementary Fig. 6).

Considering that the reduced nicotinamide adenine dinucleotide phosphate (NADPH) is critical in biosynthetic pathways serving as the electron carrier for a large subset of oxidoreductases[25], the increased lycopene yield is partially due to the enhanced NADPH in *R. palustris* by receiving electrons from the dynamic $Fe^{3+}/Fe^{2+}$ cyclic communication process through the iron oxidation pathway.(Supplementary Fig. 7)[19]. The Fe redox signal inducible verifier ($Zn_2GeO_4$:Mn) introduced into the biological LAN can not only confirm and track the relayed redox information, but also guide the optimization of biosynthesis process (Fig. 2g, Supplementary Fig. 8)[22]. Specifically, the luminescence of $Zn_2GeO_4$:Mn (ZGO:Mn) at 536 nm originates from the $^4 T_1(G) \rightarrow {}^6A_1(S)$ transition of $Mn^{2+}$ luminescence center[22]. Owing to the reduction Fermi energy level locations of $E_{Fe^{3+},red}$ and $E_{Fe^{2+},red}$, the nano verifier shows different luminescence responses toward $Fe^{3+}$ and $Fe^{2+}$, thus achieving a dynamic and reversal monitoring of the Fe redox signal conversion in real-time (Supplementary Fig. 9)[22]. In addition, the nano verifier exhibits low toxicity both against *S. putrefaciens* and *R. palustris* according to quantitative colony-forming unit (CFU) survival assays (Supplementary Fig. 10). To investigate the performance of the verifier in Fe redox signal monitoring in the biological LAN, we performed the correlation analysis between the luminescence intensity measured with the verifier method and the absorbance measured with the typical phenanthroline spectrophotometric method[26]. As shown in Fig. 2h, the

luminescence intensity of verifier is highly correlated with the measured absorbance ($R = 0.99$, $P = 0.002$), further proving that verifier can provide an accurate and dynamic luminescent monitoring of the Fe redox signal. Furthermore, based on the recorded luminescence intensity of the verifier (Supplementary Fig. 11) and the linear relationship between logarithm of persistent luminescence intensity and $Fe^{2+}/Fe_{total}$ ratio (Supplementary Fig. 9d), the concentrations of $Fe^{3+}/Fe^{2+}$ redox couple were calculated. With the calculated values, we then simulated the kinetic transformation of the $Fe^{3+}/Fe^{2+}$ redox signal according to the equations (Supplemental methods) with MATLAB software[1,27]. The simulation results in Fig. 2i show that $Fe^{3+}$ signal is transformed to $Fe^{2+}$ first and the concentration of $Fe^{2+}$ peaks around 0.76 mM, then the $Fe^{2+}$ signal decreased gradually. The simulation results of $Fe^{2+}$ concentration and $OD_{600}$ in the biological LAN are demonstrated to be highly correlated with the experimental data ($R = 0.90$, $P = 0.02$ for $Fe^{2+}$ concentration and $R = 0.99$, $P < 10^{-4}$ for $OD_{600}$), indicating that the simulation models are competent to assess or predict behaviors in the biological LAN. Overall, according to the above results, the constructed biological LAN with the ability to sense, analyze and process Fe redox signal demonstrate the achievement of social division of labor.

## Redox communication coordinates microbial metabolism

The long-term evolution of the microbial communities enables cells to sense and response to the extracellular environment[28]. According to the varied environments, microbial communities are capable to dynamically allocate cellular resources and optimally control pathway expressions, thus maintaining the stability of the microbial community[29,30]. Similarly, microbial community behavior can be self-adapted in response to the extracellular dynamic switching redox state. The confocal images in Fig. 3a show rod-shaped morphologies of *S. putrefaciens* signal router (green) and *R. palustris* bio-actuator (red) labeled by fluorescent in situ hybridization (FISH) probes[31,32]. To explore the effects of Fe redox communication on microbial function and physiology, the functional potential of the microbial community based on 16 S rRNA gene sequencing were predicted with PICRUSt2. As shown in Fig. 3b, the abundance of major function groups involved in microbial growth such as amino acids metabolism and lipid metabolism is higher, while the abundance of genes involved in energy metabolism is lower in the biological LAN with redox communication compared with the bare co-culture system. The results suggest that the introduction of Fe redox signal into the biological LAN leads to enhanced growth metabolism and reduced energy metabolism[16]. This may be due to the syntrophic cross-feeding of electrons between *S. putrefaciens* signal router and *R. palustris* bio-actuator mediated with the Fe redox signal, saving the energetic cost and increasing the excretion of critical reduced compounds, resulting in enhanced physiological functions (Supplementary Fig. 12)[33,34].

To validate the above hypothesis, the critical reduced compounds and energy rich compounds including NADPH, NADH and adenosine 5′-triphosphate (ATP) were tested. It was found that, the reducing compounds of NADPH and NADH accumulated correspondingly with the simultaneous generation of ATP in the biological LAN with redox communication (Fig. 3c–e), indicating that the electron transfer triggered redox communication could provide efficient reducing power and energy to the microbial community. In addition, it can be observed that the population composition differs between the biological LAN with redox communication and the bare co-culture system. As shown in Fig. 3f, the proportions of *R. palustris* bio-actuator at different incubation times in the biological LAN with redox communication are higher compared with the bare co-culture system (Supplementary Fig. 13). The $OD_{600}$ values of *R. palustris* bio-actuator are about 2.0-fold higher compared with the values in bare co-culture system across time (Fig. 3g), while the $OD_{600}$ values of *S. putrefaciens* are similar with the

values in bare co-culture system across time (Supplementary Fig. 14). The above results indicate that the redox communication process both modulates the metabolism behaviors and the population compositions of the microbial community.

## Redox communication regulates the gene expression of electron transfer related proteins

To have a further understanding of the redox communication mechanism in the biological LAN, transcriptomic analysis of *S. putrefaciens* signal router and *R. palustris* bio-actuator was performed[35,36]. Differentially expressed genes were evaluated according to the $Log_2$ fold change ($Log_2$ FC), which was calculated with {$Log_2$ [(Gene read counts)$_{(Experimental)}$] – $Log_2$ [(Gene read counts)$_{(Control)}$]}. Overrepresented (white to purple for *S. putrefaciens* and white to red for *R. palustris*) and underrepresented transcript (white to green for *S. putrefaciens* and white to blue for *R. palustris*) with the reference of gene expressions in bare co-culture shown as $Log_2$ FC were exhibited in Fig. 4a. It can be observed that the $Log_2$ FC of electron transfer proteins including flagellum, pilus and c-type cytochromes (c-Cyts) and PioABC complex are >0 on the whole, indicating that genes encoding electron transfer proteins were actively expressed in the biological LAN with redox communication. In addition, genes associated with reducing power substance also demonstrated high expression in the biological LAN with redox communication. The highly expressed genes typically represent the highly express of related proteins. Accordingly, it is suggested that the redox communication might improve the functions of electron transfer proteins, thus fascinating the electron transfer process between *S. putrefaciens* signal router and *R. palustris* bio-actuator. To validate the increased electron transfer function, current density of cells was measured with gold interdigitated microelectrode (Supplemental methods). The current density results demonstrate that cells in the biological LAN with redox communication have about 2.10 times higher current density compared with cells without redox communication, suggesting the improved electron transfer capability (Supplementary Fig. 15). Besides, Fig. 4a also shows that the $Log_2$ FC of proteins associated with both the MEP & MVA pathways and $CO_2$ fixation are above 0 generally, suggesting that the redox communication triggered with electron transfer facilitates the biosynthesis and $CO_2$ fixation processes in microorganism (Supplementary Fig. 16).

Specifically, of 4562 transcripts detected in *R. palustris*, 169 were significantly associated with the redox communication (*P value* < 0.05 and FC ≥ |1.5|), indicating that the redox communication network influences the gene expressions of microorganism (Fig. 4b). The conclusion was further validated according to differentially expressed genes analysis in *S. putrefaciens* (Supplementary Fig. 17). Subsequently, by assessing the mRNA expressions of *R. palustris* and *S. putrefaciens*, it was found that the electron transfer and reducing power associated genes had higher expression level represented as $Log_2$ FPKM value in the biological LAN with redox communication compared with the bare co-culture (Fig. 4c, d, Supplementary Fig. 18). The above results demonstrate that the Fe signal transduction in biological LAN could upregulate gene expressions of electron transfer and reducing power generation related proteins, thus regulating microbial metabolic behaviors.

## Redox communication facilitates the biomanufacturing efficiency of *R. palustris* bio-actuator

The essential of the redox state dynamic conversion is the circulation flow of electrons. In the biological LAN with redox communication, the dynamic conversion of $Fe^{3+}/Fe^{2+}$ redox couple is able to channel electrons directly to *R. palustris* bio-actuator through the iron oxidation pathway (Supplementary Fig. 19)[19,20,37]. Specifically, PioA and PioB are proposed to oxidize $Fe^{2+}$ extracellularly and then transfer the released

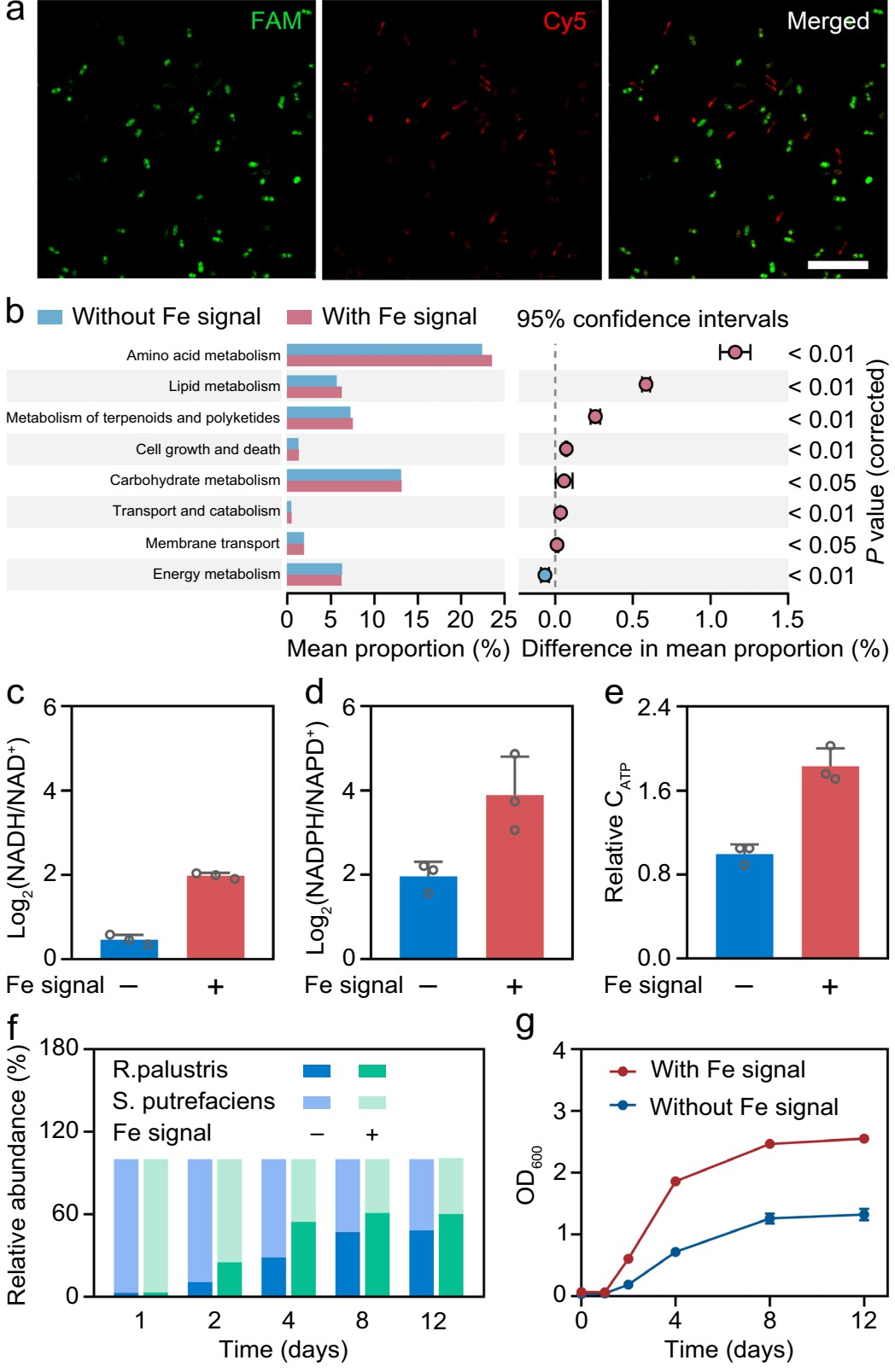

**Fig. 3 | Microbial behaviors in the biological LAN with redox communication.**
**a** Confocal fluorescence imaging of *S. putrefaciens* (FAM, green) and *R. palustris* (Cy5, red) in the biological LAN labelled with the FISH probes. Scar bar is 10 μm. At least three independent experiments were carried out with similar results. **b** Gene function prediction of *S. putrefaciens*-*R. palustris* co-culture in the biological LAN by PICRUSt2. The middle value represents the mean differences between the groups, and the error bar represents the 95% confidence intervals. **c**–**e** Log₂ (NADH/NAD⁺) ($c$), Log$_2$ (NADPH/NADP⁺) (**d**) ratio and relative ATP concentration (**e**) of *S. putrefaciens*-*R. palustris* co-culture in the biological LAN incubated at 96 h. Data presented as mean values ± SD, $n = 3$. **f** Relative abundance of *S. putrefaciens* and *R. palustris* over incubation time in the biological LAN. **g** Calculated OD$_{600}$ of *R. palustris* in the biological LAN. Data presented as mean values ± SD, $n = 3$. Source data are provided as a Source Data file.

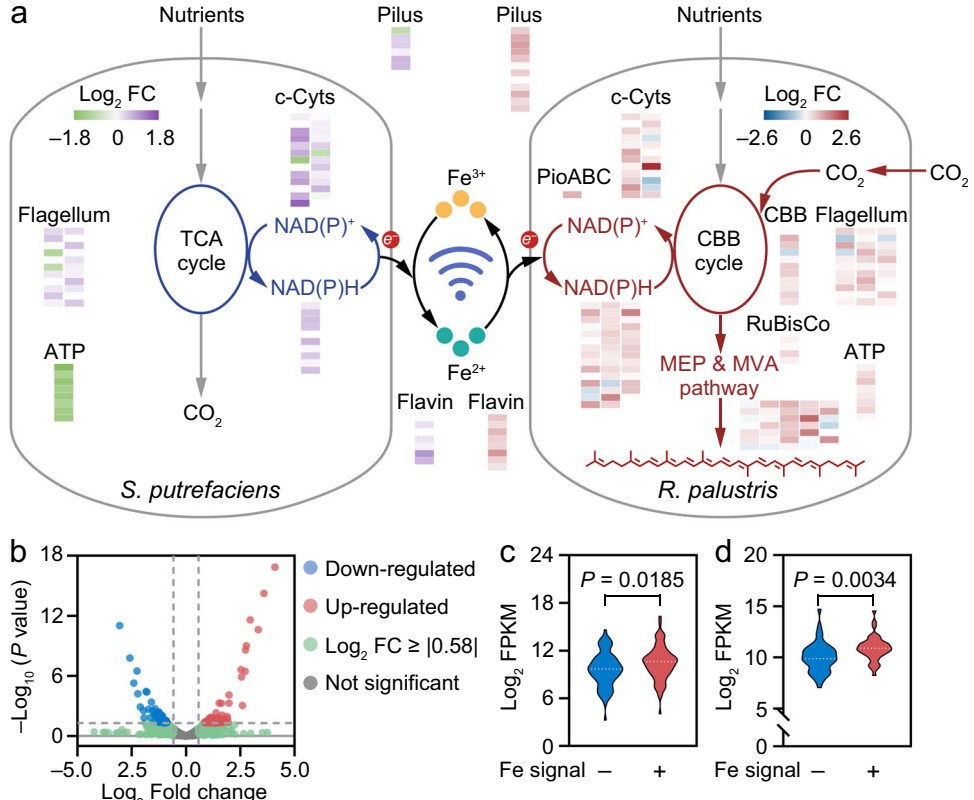

**Fig. 4 | Differential gene expression analysis of *S. putrefaciens* and *R. palustris* in the biological LAN. a** Schematic illustration of electron cross feeding between *S. putrefaciens* and *R. palustris* in the biological LAN with Fe redox communication. Inset heat maps show the differential gene expression presented as $Log_2$ FC value. **b** A Volcano plot of differentially expressed genes in *R. palustris*. The threshold of $Log_2$ FC is |0.58| (i.e., FC ≥ |1.5|), and that of *P* value is <0.05. There were 169 differentially expressed microbial transcripts that met these criteria. **c,d** Violin plots showing the significant differences in expression of (**c**) electron transfer related genes (including pilus, c-Cyts, flagellum, PioABC, flavin) and (**d**) reducing equivalent related genes of *R. palustris* (including NAD(P)H related proteins). *P* values were determined by an unpaired two-sided *t*-test. Source data are provided as a Source Data file.

electrons across the outer membrane to PioC, which is hypothesized to be in the periplasm[19]. Previous studies have proved that the oriented electron transfer is critical in promoting reducing power NADPH for high biomanufacturing efficiency[16,38,39]. Hence, we sought to investigate the influence of directed electron flow on lycopene biosynthesis yields of *R. palustris* bio-actuator in our constructed biological LAN.

Carbonyl cyanide m-chlorophenyl hydrazone (CCCP) is a lipid-soluble molecule and has been reported to uncouple the electron transfer from ATP synthesis (Supplementary Fig. 20). Rotenone, previously reported as an inhibitor for NADH dehydrogenase, is capable to block electron transfer from the iron-sulfur clusters in NADH dehydrogenase to ubiquinone (Supplementary Fig. 21)[40]. Both CCCP and rotenone are recognized to inhibit the electron transfer process in *R. palustris* bio-actuator[40]. It is observed that upon CCCP and rotenone treatment, the lycopene biosynthesis yields of *R. palustris* bio-actuator in the biological LAN with redox communication decreased 40.64% and 40.70% respectively compared with the untreated controls (Fig. 5a), demonstrating that the hindered electron transfer processes decrease the biosynthesis efficiency of *R. palustris* bio-actuator. Furthermore, upon the addition of electron shuttle of humic acid (HA) and anthraquinone-2,6-disulfonate (AQDS), the biosynthesis yields increase 54.05% and 88.86% respectively in the redox-based communication network, suggesting that accelerated electron transfer processes increase the biosynthesis efficiency (Fig. 5b). Similarly, the biosynthesis efficiency can be obviously improved through the addition of electron donors for microbial respiration including acetate and lactate[20,41] (Supplementary Fig. 22). The observations intriguingly suggested that the

electron transfer process may indeed be involved in regulating biosynthesis metabolism in *R. palustris* bio-actuator.

To validate the relationship between electron transfer and biosynthesis metabolism in *R. palustris* bio-actuator, fluorescence-activated cell sorting (FACS) was performed to isolate the *R. palustris* bio-actuator from *S. putrefaciens-R. palustris* co-culture for further metabolism investigation. As shown in Fig. 5c, d, the values of $Log_2$ ratio of NADH/NAD+ (2.94 ± 0.24) and NADPH/NADP+ (3.71 ± 0.62) in *R. palustris* bio-actuator with Fe redox communication are higher than the values without Fe redox communication, largely due to the acceptance of external electrons of $Fe^{2+}$ with the iron oxidation pathway in *R. palustris*[22,25]. Accordingly, as schemed in Fig. 5e, electrons can be supplied continuously and delivered specifically into *R. palustris* bio-actuator through the dynamic and reversible $Fe^{3+}/Fe^{2+}$ signal transduction process in the redox-based communication network, allowing to enhance the generation of reducing power NADPH. It is widely recognized that NADPH acts as critical electron carrier for a large subset of oxidoreductases to drive the microbial biosynthesis metabolism[25]. Hence, the lycopene biosynthesis yields can be improved through the MVA and MEP pathway in *R. palustris* with the continuous supply of electrons. Subsequently, fed-batch fermentation at laboratory scale was further performed to explore the applicability of the proposed redox based communication strategy. It is inspiring to find that the lycopene biosynthesis increased about 1.43 times in fed-batch fermentation compared with the shake-flask fermentation (Supplementary Fig. 23, Supplementary Methods). The above results indicated the potential application prospects of the proposed redox based communication strategy in practical lycopene biomanufacturing.

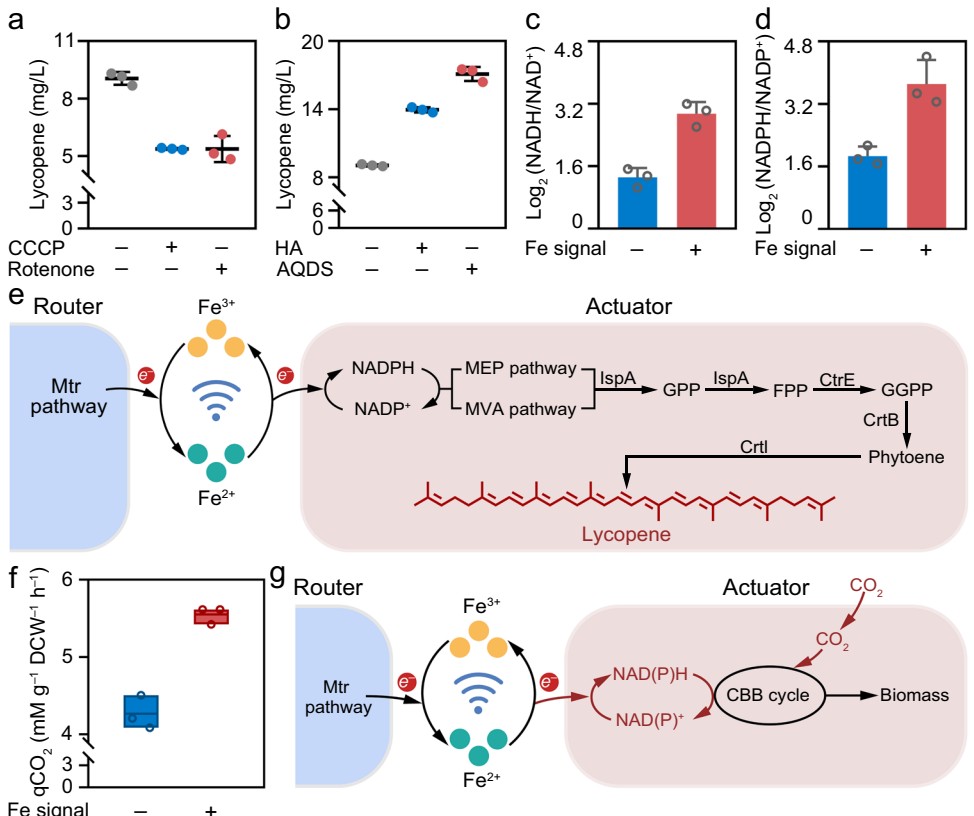

**Fig. 5 | Biomanufacturing function of *R. palustris* in the biological LAN.**
**a** Lycopene yields of *R. palustris* in the Fe redox based biological LAN with or without the addition of electron transfer related protein inhibitor. Data presented as mean values ± SD, $n = 3$. **b** Lycopene yields of *R. palustris* in the Fe redox based biological LAN with or without the addition of electron transfer shuttle. Data presented as mean values ± SD, $n = 3$. **c,d** $Log_2$ (NADH/NAD$^+$) (**c**) and $Log_2$ (NADPH/NADP$^+$) (**d**) of the isolated *R. palustris* from the biological LAN. Data presented as

mean values ± SD, $n = 3$. **e** Schematic illustration of lycopene biomanufacturing process of *R. palustris* in the Fe redox based biological LAN. **f** $CO_2$ fixing rate of *R. palustris* in the Fe redox based biological LAN. Data presented as mean values ± SD, $n = 3$. Box edges indicate the upper and lower quartiles, the centerlines indicate the mean value, the horizontal bars indicate the maximum and minimum values. **g** Schematic illustration of $CO_2$ fixation of *R. palustris* in the Fe redox based biological LAN. Source data are provided as a Source Data file.

*R. palustris* bio-actuator is a typical photosynthetic bacterium in which $CO_2$ can be fixed through the Calvin-Benson-Bassham (CBB) cycle. $CO_2$ fixation process requires the investment of reducing power[36,42,43]. The constructed redox-based communication network has demonstrated to be able to supply electrons and subsequently accelerate reducing power in *R. palustris*. The possibility that redox communication process could drive the reducing power generation and therefore improving the efficiency of $CO_2$ fixation was further investigated. As shown in Fig. 5f, the $CO_2$ fixation rate of *R. palustris* in redox-based communication network reaches 5.55 Mm g$^{-1}$ DCW$^{-1}$ h$^{-1}$, 1.30 times higher compared with the rate in bare co-culture, validating that the redox communication network improves efficiency of $CO_2$ fixation in *R. palustris*. The enhanced $CO_2$ fixation rate might be due to the directed electron transfer provided by the dynamic Fe$^{3+}$/Fe$^{2+}$ cyclic process in the redox communication network, affording the critical reducing power for CBB cycle (Supplementary Fig. 24).

### Redox communication in microbial consortia
Microbial consortia are highly desired for efficient biomanufacturing with the distinct advantages of alleviating metabolic burden by division of labor through trading metabolites or exchanging signals and has recently attracts considerable interest[44,45]. The applicability of the redox communication strategy was further investigated in microbial consortia. As schemed in Fig. 6a, a Fe redox based biological LAN consisted of dual-band router, optical verifier and bio-actuator was constructed (Supplementary Fig. 25). Similar with *S. putrefaciens*, *Geobacter soli* (*G. Soli*) is also an iron-reducing bacterium that can

convert Fe$^{3+}$ redox signals into Fe$^{2+}$. The dual-band router composed of *S. putrefaciens* and *G. Soli* can ensure the stably Fe$^{2+}$ signal output according to their respective metabolic state, avoiding the occurrence of "unstable network" or "disconnected" situations[46].

As shown in Fig. 6b, the verifier exhibited increased luminescence intensity in *S. putrefaciens* and *G. Soli* co-culture compared with the bare *S. putrefaciens* and bare *G. Soli*, suggesting that the dual-band router is competent for efficient Fe$^{2+}$ signal output (Supplementary Fig. 26). In addition, with Fe redox communication, the lycopene biosynthesis yield and $CO_2$ fixation rate of *R. palustris* bio-actuator is 1.55 and 1.48 times higher respectively compared with the bare microbial consortia (Fig. 6d, e). The promoted biosynthesis and $CO_2$ fixation efficiency is largely due to the directed and specific electron transfer from Fe$^{2+}$ to *R. palustris* bio-actuator, leading to the accumulation of critical reducing power for metabolism (Fig. 6f, g). The above results demonstrate that the electron transfer triggered redox communication strategy is also capable to improve the biosynthesis and biological $CO_2$ fixation efficiency in microbial consortia. The designed electron transfer triggered redox communication network proposes a general strategy for microbial metabolism coordination and regulation, offering insights into reducing power regeneration and metabolic optimization.

## Discussion
Microbial communication, capable to drive coordinated functions through sensing, analyzing and processing signal information, plays critical roles in biomanufacturing and life evolution, and has attracted

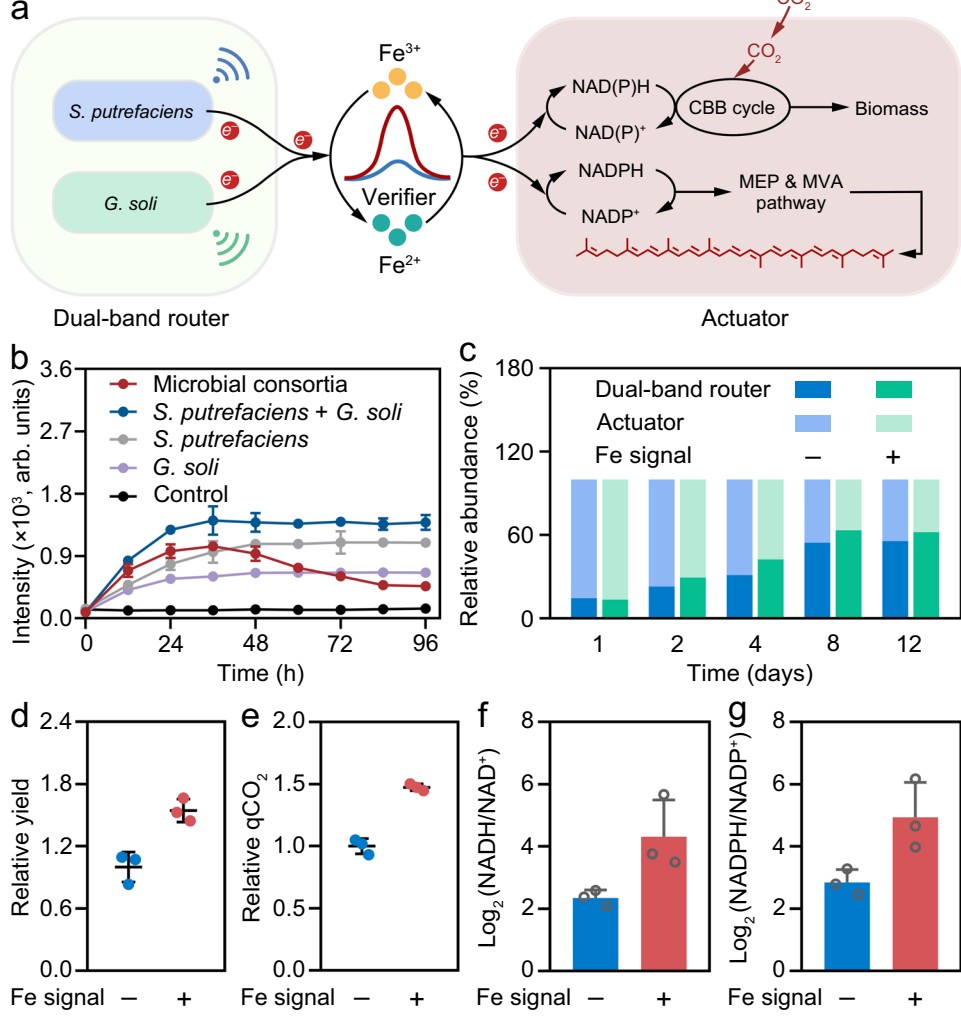

**Fig. 6 | Redox communication in microbial consortia. a** Schematic illustration of the biological LAN consisted of dual-band router, optical verifier and bio-actuator. **b** The persistent luminescence of verifier in the microbial consortia. **c** Relative abundance of bacteria in the microbial consortia. Data presented as mean values ± SD, $n = 3$. **d** Relative lycopene biomanufacturing yield of *R. palustris* bio-actuator in the microbial consortia with or without Fe redox communication.

Data presented as mean values ± SD, $n = 3$. **e** Relative $CO_2$ fixing rate of *R. palustris* bio-actuator in the microbial consortia with or without Fe redox communication. Data presented as mean values ± SD, $n = 3$. **f,g** $Log_2$ (NADH/NAD$^+$) (**f**) and $Log_2$ (NADPH/NADP$^+$) (**g**) ratio of the isolated *R. palustris* from the microbial consortia. Data presented as mean values ± SD, $n = 3$. Source data are provided as a Source Data file.

considerable recent interest. Redox-based communication provides an expedient tool for intelligent self-adaptive regulation and is promising in metabolism coordination. Much pioneering work has been performed on redox-based communication for guiding biological function[1,13]. For example, Bentley and the co-workers have long engaged in bio-electronic control of microbial metabolism with electrogenetic device that uses redox biomolecules to carry electronic information to engineered bacterial cells[1,13]. From another perspective, this work proposed a redox-based communication network strategy that was independent of external electronics input and the intrinsic electron transfer triggered redox cycle could function as batteries that facilitate the biosynthesis processes. Our study provides fundamental insights into the redox communication between microbial species and revealed the metabolism coordination mechanism. A biological local area network model consisted of three building blocks including signal routing cell (*S. putrefaciens*), optical verifier and bio-actuation cell (*R. palustris*) based on electron transfer triggered Fe redox communication strategy was constructed as a proof of concept. It is demonstrated that the redox communication network drives gene expressions of electron transfer proteins and simultaneously facilitates the critical reducing power regeneration in "bio-actuator cells", enabling

microbial metabolism regulation. In this way, the biomanufacturing yield of lycopene in "bio-actuator cells" increased with the mevalonate and methylerythritol 4-phosphate pathway. In addition, our work demonstrated that the proposed redox communication strategy is capable to facilitate $CO_2$ fixation and is applicable both in co-culture and microbial consortia. The demonstration of microbial behavior coordination with the electron transfer triggered redox communication strategy represents a critical step to construct robust metabolism regulation platform for engineering microbial metabolic relevant activities. Looking forward, with the development of synthetic biology, we believe that the proposed redox communication strategy may pave ways for self-adaption and dynamic microbial metabolism regulation and is promising in facilitating the development of mixed fermentation industries with the distinct advantages of alleviating metabolic burden by division of labor.

## Methods
### Materials
Germanium oxide (GeO$_2$, 99.99%), zinc nitrate hexahydrate (Zn(NO$_3$)$_2$·6H$_2$O), manganese nitrate (Mn(NO$_3$)$_2$), manganese powder, manganese(II) chloride (Mn(Cl)$_2$), manganese dioxide (MnO$_2$), iron (III)

chloride hexahydrate ($FeCl_3 \cdot 6H_2O$), iron(II) chloride ($FeCl_2$), O-Phenanthroline were purchased from Sinopharm Chemical Reagent Co., Ltd. Concentrated nitric acid ($HNO_3$), concentrated sulfuric acid ($H_2SO_4$), sodium hydroxide (NaOH), lactate, sodium phosphate dibasic heptahydrate ($Na_2HPO_4 \cdot 7H_2O$), potassium dihydrogen phosphate ($KH_2PO_4$), sodium chloride (NaCl), ammonium chloride ($NH_4Cl$), sodium fumarate, humic acid (HA), anthraquinone-2,6-disulfonate (AQDS), acetonitrile, methanol, chloroform, glutathione disulfide (GSSG) and glutathione (GSH) were purchased from Aladdin Biochemical Technology Co., Ltd. Hydroxylamine hydrochloride, sodium acetate, acetic acid, ammonium iron (II) sulfate ($Fe(NH_4)_2 \cdot (SO_4)_2 \cdot 6H_2O$) were purchased from Macklin Biochemical Co., Ltd. Glycerin, protonophore carbonyl cyanide m-chlorophenyl hydrazone (CCCP), rotenone, formamide were purchased from Sigma-Aldrich Trading Co., Ltd. Peptone, yeast extract, paraformaldehyde, 10% SDS solution, agar, PBS buffer (PH = 7.2–7.4), Tris-HCl buffer solution (PH = 7.5) were purchased from Solarbio Science & Technology Co., Ltd. The bacteria strains *S. putrefaciens* (BNCC 337021) and *R. palustris* (BNCC 232041) were obtained from Bnbio Co., Ltd. *G. Soli* (CCTCC AB 2014145) was provided by China Center for Type Culture Collection. The FISH probes for *S. putrefaciens*, *R. palustris* and *G. Soli* with the sequences of 5′-FAM- AGCTAATCCCACCTAGGTTCATC-3′, 5′-FAM-AGCTAATCCCACCTAGGTTCATC-3′ were synthesized by Shanghai Sangon Biotech Co., Ltd. $NADP^+$/NADPH Assay Kit with WST-8, $NAD^+$/NADH Assay Kit with WST-8 and Enhanced ATP Assay Kit were purchased from Beyotime Biotech. Inc. The ultra-pure water was obtained using a Millipore water purification system.

## Characterization

The morphology of ZGO:Mn nano verifier was measured by a transmission electron microscope (TEM) (JEOL, JEM-2100, Japan) with a working voltage of 200 kV. Powder X-ray diffraction (XRD) patterns of ZGO:Mn nano verifier was measured on an X-ray diffractometer (Burker, D8 Advance, Germany) with Cu-Kα radiation (λ = 1.5406 Å) to determine the crystal structure. Absorbance was measured and recorded on a BioTeK Synergy H1 Hybrid Reader (BioTek, USA). Persistent luminescence performance of ZGO:Mn nano verifier was measured on a Hitachi FL4600 fluorescence spectrometer (Hitachi, Japan). The morphologies of *S. putrefaciens* and *R. palustris* were observed with a field emission scanning electron microscope (SEM) (Zeiss, SIGMA,Germany). Microbial consortia were imaged on the Olympus Fluoview FV3000 confocal microscope (Olympus, Japan). Current intensity of microbial cells was measured by a digital source meter (Keysight B1500A, Germany) connected to a probe station (PRCBE LAB, China). Fluorescence-activated cell sorting (FACS) was performed by BD FACSAria III Cell Sorter (BD Biosciences, USA). The Agilent 1260 Infinity HPLC system (Agilent, USA) was used to measure the concentration of lycopene.

## Scanning electron microscopy characterization

The morphologies of *S. putrefaciens* signal router and *R. palustris* bioactuator were observed with a field emission scanning electron microscope (Zeiss, Germany). Specifically, bacteria solution with $OD_{600}$ of 0.6 was centrifuged and then washed twice with ultra-pure water. The washed bacteria were fixed in 10 mL 4% paraformaldehyde for 1.5 h. After the fixation process, the bacteria were washed twice with the rotation speed of 4000 rpm. Subsequently, 5 mL of 25, 50, 75, 100% ethanol solution was used in order to dehydrate the bacteria. Each dewatering time was 15 min. Finally, the dehydrated bacteria were centrifuged and dissolved in anhydrous ethanol for preparing SEM samples.

## Performance measurements of the ZGO:Mn nano verifier

To investigate the relationship between the persistent luminescence intensity and Fe species concentration, standard solution with different $Fe^{2+}$/$Fe_{total}$ ratio was prepared with $FeCl_2$ and $FeCl_3 \cdot 6H_2O$ ($Fe_{total}$ was

fixed at 2 mM, and the final concentration of ZGO:Mn nano verifier was 0.5 mg/mL). Then, the persistent luminescence intensity of the different solution was measured with a fluorescence spectrometer (Hitachi, FL4600, Japan). The excitation wavelength was 254 nm and the data mode was set as phosphorescence. Similarly, we measured the persistent luminescence intensity upon the addition of Mn (0.110 mg/mL), $MnCl_2$ (2 mM), $MnO_2$ (0.174 mg/mL), GSSG (2 mM), GSH (2 mM), supernatant and medium with fluorescence spectrometer (Hitachi, FL4600, Japan).

## Monitoring of the Fe species in biological LAN with the nano verifier

To have a better understanding of the Fe signal transduction between *S. putrefaciens* and *R. palustris*, ZGO:Mn nano verifier was adopted. Specifically, ZGO:Mn with final concentration of 0.5 mg/mL was added into 5 mL sterilized tubes containing 2 mL LB medium and 2 mM $Fe^{3+}$. Subsequently, 20 uL *S. putrefaciens* and 20 uL *R. palustris* with the $OD_{600}$ around 0.8 was added. Then, the centrifuge tubes with mixed bacteria solution were placed into a shaker with a speed of 220 rpm at 30 °C. The persistent luminescence intensity of the mixed solution was measured at different culture time with a fluorescence spectrometer (Hitachi, FL4600, Japan). The excitation wavelength was 254 nm and the data mode was set as phosphorescence.

## Cell viability tests

The colony-forming unit (CFU) was performed to determine the viability of *S. putrefaciens* signal router and *R. palustris* bio-actuator after incubation with the optical ZGO:Mn nano verifier. Specifically, diluted *S. putrefaciens*-ZGO:Mn and *R. palustris*-ZGO:Mn co-culture solution at different culture times was transferred into the Luria-Bertani (LB) solid medium respectively. After 24 h growth at 30 °C, white circular colonies can be observed. The CFU per milliliter can be determined by counting the colonies.

## Fe signal transduction and bacterial growth model in biological LAN

Fe redox signal transduction processes in the biological LAN were modelled with Simulink 9.7 (R2019b). The systems of ordinary differential equations below were solved using ode45 solver. The mathematical model here can be used to investigate in silico the redox signal transduction kinetic behaviors in continuous cultures. In the constructed model, it is supposed that there are no delays in responses to signals to simplify the actual system. In addition, 1 OD was assumed to be $10^9$ cells. The model parameters were calculated using a combination of the experimental data and the model.

Specifically, the concentrations of $Fe^{3+}$ and $Fe^{2+}$ were calculated based on the recorded luminescence intensity of the verifier according to the following equation (Supplementary Fig. 9d) in which $Fe_{Total}$ was 2 mM:

$$Lg(Intensity) = 0.01764 * C_{Fe^{2+}/Fe_{Total}} + 2.126 \tag{1}$$

Then, based on the calculated concentrations, the $Fe^{2+}$ signal transduction from $Fe^{3+}$ in bare *S. putrefaciens* system was simulated with Eq. (2) to fit and abstain the constants:

$$\left[Fe^{2+}\right]_{S'} = B_S + \frac{A_S - B_S}{1 + \left(\frac{t}{C_S}\right)^{D_S}} \tag{2}$$

Similarly, the $Fe^{3+}$ signal transduction from $Fe^{2+}$ in bare *R. palustris* system was simulated with Eq. (3) to fit and abstain the constants:

$$2 - \left[Fe^{2+}\right]_{R'} = B_R + \frac{A_R - B_R}{1 + \left(\frac{t}{C_R}\right)^{D_R}} \tag{3}$$

Based on the above models of the bare *S. putrefaciens* and bare *R. palustris* system, the redox signal transduction and bacterial growth models of the co-culture system were established. Considering that *S. putrefaciens* transduces $Fe^{3+}$ signal into $Fe^{2+}$ while the $Fe^{2+}$ can be transduced back to $Fe^{3+}$ by *R. palustris*, a redox cycle module was introduced in the co-culture model to simulate Fe redox communication. The initial $Fe^{3+}$ concentration was set at 2 mM which is consistent with the experimental data. In addition, Fe redox communication influence on bacterial growth was also considered. Overall, the differential equations were obtained as bellowing:

*S. putrefaciens* density in co-culture system:

$$\frac{d[Cells]_S}{dt} = \left[Fe^{2+}\right]_S * [Cells]_S * \frac{E_S}{F_S + \frac{G_S - F_S}{1 + \left(\frac{t}{H_S}\right)^{I_S}}} \tag{4}$$

*R. palustris* density in co-culture system:

$$\frac{d[Cells]_R}{dt} = \left(2 - \left[Fe^{2+}\right]_R\right) * [Cells]_R * \frac{E_R}{F_R + \frac{G_R - F_R}{1 + \left(\frac{t}{H_R}\right)^{I_R}}} \tag{5}$$

$Fe^{2+}$ concentration in co-culture system:

$$\frac{d\left[Fe^{2+}\right]}{dt} = \frac{d\left[Fe^{2+}\right]_S}{dt} - \frac{d\left[Fe^{2+}\right]_R}{dt} = [Cells]_S * \frac{J_S}{K_S + \frac{L_S - K_S}{1 + \left(\frac{t}{M_S}\right)^{N_S}}}$$
$$- [Cells]_R * \frac{J_R}{K_R + \frac{L_R - K_R}{1 + \left(\frac{t}{M_R}\right)^{N_R}}} \tag{6}$$

## Bacteria culture

*S. putrefaciens*, *R. palustris*, *S. putrefaciens-R. palustris* co-culture and *S. putrefaciens-R. palustris-G. Soli* microbial consortia were all cultured in Luria-Bertani (LB) medium that contains (per litre) 10.0 g peptone, 5.0 g yeast extract and 5.0 g NaCl. *G. Soli* was cultured in $N_2$-flushed LM medium that contains (per litre) 2.0 g lactate, 0.2 g yeast extract, 12.8 g $Na_2HPO_4 \cdot 7H_2O$, 3 g $KH_2PO_4$, 0.5 g NaCl and 1.0 g $NH_4Cl$. Additional sodium acetate (20 mM) as the electron donor and sodium fumarate (50 mM) as the electron acceptor were added into the above LM medium. All the bacteria were cultured at 30 °C with a shaking rate of 220 rpm. *R. palustris* was photosynthetic bacteria and a full-spectrum LED (100 W) was used as its light source.

## Preparation of the optical ZGO:Mn verifier

The optical ZGO:Mn verifier was synthesized by a hydrothermal method. Typically, a mixture solution was obtained by dissolving 0.01 mmol $Ga(NO_3)_3$, 2 mmol $Zn(NO_3)_2$, 0.005 mmol $Mn(NO_3)_2$ and 300 μL $HNO_3$ into in 11 mL deionized water. Then 1 mmol $Na_2GeO_3$ solution was prepared by dissolving $GeO_2$ powder in 2 mol $L^{-1}$ NaOH solution. Subsequently, the prepared $Na_2GeO_3$ solution was added drop by drop into the above mixture solution. Ammonium hydroxide (28 wt%) was used to adjust the pH value of the mixture solution to 9.5. Then, the reaction was left at room temperature under stirring for 1 h. After that, the solution was transferred into a Teflon-lined autoclave and reacted at 220 °C for 6 h. The resultant ZGO:Mn nanorods were then obtained by centrifugation and washed three times with deionized water.

## Construction of redox based biological LAN

The initial inoculation of *S. putrefaciens* and *R. palustris* were cultured in LB medium. To construct redox based biological LAN, *S. putrefaciens* signal router, optical verifier and *R. palustris* bio-actuator were

introduced. Specifically, 1% of the seed culture of *S. putrefaciens* and *R. palustris* with initial $OD_{600}$ around 0.8 were transferred respectively into sterilized centrifuge tubes with LB medium. Then, optical ZGO:Mn verifier (0.5 mg/mL) and $Fe^{3+}$ (2 mM) were added. The biological LAN including the *S. putrefaciens-G. Soli* dual-band router was constructed as the above procedure except for the addition of *G. Soli*.

## Redox signal verification in the biological LAN

The redox signal transduction can be monitored in real-time with the optical ZGO:Mn verifier. Briefly, the persistent luminescence intensity of the mixed solution at different time was measured with a Hitachi FL4600 fluorescence spectrometer (Hitachi, Japan). The excitation wavelength was 254 nm and the data mode was set as phosphorescence.

## Construction of ΔCymA and ΔMtrC mutants

Mutants with MtrC-encoding gene deleted (ΔMtrC) and CymA-encoding gene deleted (ΔCymA) were constructed respectively as previous report. Specifically, an upstream and downstream fragment were amplified separately from the *S. putrefaciens* genomic DNA by PCR using the primer pair Mtrc-UP-smaI-F/Mtrc-UP-R, and the pair Mtrc-down-F/Mtrc-down-kpnI-R, respectively (Table S1). Then, the PCR products were purified using a QIAquick Gel Purification Kit (QIAGEN Inc.) and cloned into the SmaI/KpnI sites of pRE112 vector. Subsequently, the recombinant plasmid transformed in *E. coli* S17-1/λpir was transferred into *S. putrefaciens* via the *E. coli-Shewanella* conjugation. Finally, the Mtrc deletion mutants were screened and its genotype was confirmed by PCR using the primer pair Mtrc-JD-F/Mtrc-JD-R (Table S1) and DNA sequencing. Similarly, the CymA deletion mutants can be constructed according to the above protocol with its own primer pair (Table S1).

## o-Phenanthroline spectrophotometric method for $Fe^{2+}$ quantification

The well-known o-phenanthroline spectrophotometric method was adopted to quantify the concentration of $Fe^{2+}$. Typically, 0.7 g $Fe(NH_4)_2 \cdot (SO_4)_2 \cdot 6H_2O$ was dissolved in 1 M $H_2SO_4$ solution to prepare the $Fe^{2+}$ standard solution (100 mg·$L^{-1}$). 20 μL 10% (w/w) hydroxylamine hydrochloride solution, 50 uL of o-Phenanthroline (0.5 wt %) and 50 uL of sodium acetate buffer (pH = 5) were added into a 96-well plate. Subsequently, 0, 2, 4, 6, 8, 10, and 12 μL $Fe^{2+}$ standard solution was added respectively into the 96-well plate. Each well was fixed to 200 μL with ultra-pure water. The above 96-well plate was left at room temperature for 15 min and the absorbance was measured at 510 nm to obtain the standard curve. Subsequently, the $Fe^{2+}$ concentrations of different samples were measured. Specifically, 50 uL of o-Phenanthroline (0.5 wt%) and 50 uL of sodium acetate buffer (pH = 5) were added into a 96-well plate. Then, sample solution was added into the 96-well plate. Each well was fixed to 200 μL with ultrapure water. The above 96-well plate was left at room temperature for 15 min and the absorbance was measured at 510 nm. Hence, the concentration of $Fe^{2+}$ can be quantified according to the obtained standard curve.

## Confocal fluorescence microscopy imaging

*S. putrefaciens-R. palustris* co-culture and *S. putrefaciens-R. palustris-G. Soli* microbial consortia were imaged with a Olympus Fluoview FV3000 confocal microscope (Olympus, Tokyo, Japan). Specifically, 10 mL cultured bacteria were first washed twice and resuspended in 4% paraformaldehyde solution for 1.5 h to fix the bacteria. The samples were then washed twice with PBS and resuspended in ethanol-PBS (V/V = 1:1) solution. Subsequently, the samples were storage at −20 °C for over 24 h. Subsequently, the samples were washed twice and resuspended in a hybridization buffer (30% formamide, 0.01% SDS, 20 mM

Tris and 0.9 M NaCl). The FISH probes of *S. putrefaciens* (5′-FAM-AGCTAATCCCACCTAGGTTCATC-3′) and *R. palustris* (5′-Cy5-CCTCTGACTTAGAAACCCGC-3′) were then added. Both the FISH probes had a final concentration of 5 μg/mL. The mixed solution was then incubated overnight at 50 °C. After the hybridization, the samples were washed with washing buffer (0.01% SDS, 20 mM Tris and 0.9 M NaCl). Finally, the samples were prepared for the confocal imaging.

## Fluorescence-activated cell sorting

As described above, fixed bacterial samples of *S. putrefaciens-R. palustris* co-culture and *S. putrefaciens-R. palustris-G. Soli* microbial consortia were resuspended in a hybridization buffer (30% formamide, 0.01% SDS, 20 mM Tris and 0.9 M NaCl). Then, The FISH probe of *R. palustris* (5′-Cy5-CCTCTGACTTAGAAACCCGC-3′) was added into the above prepared solution with a final concentration of 5 μg/mL. The mixed solution was then incubated overnight at 50 °C. After the hybridization, the samples were washed with washing buffer (0.01% SDS, 20 mM Tris and 0.9 M NaCl). Finally, the samples were prepared for flow cytometry to isolate the *R. palustris* bio-actuator cells.

## Lycopene extraction and titration

The lycopene yields at different conditions were measured. Specifically, the bacteria cells cultured for 96 h at different conditions were harvested by centrifugation at $5000 \times g$ for 5 min. The collected bacteria were washed twice with 1.0 g L$^{-1}$ NaCl. The cells were dried at 70 °C overnight to remove the water. Subsequently, lycopene in *R. palustris* bio-actuator was extracted with 3 mL mixed reagent of n-hexane and methanol (1:1 v/v). The mixed solution was vortexed for 5 min and then centrifuged at $7500 \times g$ for 10 min at 4 °C. Finally, the exacted lycopene in supernatant was collected. The concentration of lycopene was measured by the Agilent 1260 Infinity II HPLC system (Agilent, Palo Alto, CA, USA). This involved a C18 column ($4.6 \times 100$ mm, Agilent) with the mobile phase consisted of acetonitrile/methanol/chloroform (42.5:42.5:15 v/v/v). The flow rate was set at 1.0 mL min$^{-1}$ and the injection amount was set at 10 uL. The final concentrations of CCCP, rotenone, AQDS and HA used in this manuscript are 1 μM, 100 μM, 250 μM and 0.5 mg/mL respectively.

## Determination of cellular redox state and ATP concentration

*R. palustris* bio-actuator was isolated from the *S. putrefaciens-R. palustris* co-culture and *S. putrefaciens-R. palustris-G. Soli* microbial consortia through flow cytometry by labeling FISH probe of *R. palustris* (5′- Cy5-CCTCTGACTTAGAAACCCGC-3′). The collected *R. palustris* bio-actuator through fluorescence-activated cell sorting were centrifuged and washed with PBS solution (0.01 M, PH = 7.2−7.4). In particular, NAD$^+$/NADH and NADP$^+$/NADPH ratios were measured with purchased NAD$^+$/NADH Assay Kit with WST-8 and NADP$^+$/NADPH Assay Kit with WST-8. According to the protocol, the NAD$^+$/NADH and NADP$^+$/NADPH ratios were determined with a colorimetric assay under a microplate reader by detecting wavelength of 450 nm. For ATP concentration measurements, an Enhanced ATP Assay kit was adopted. The ATP concentration can be obtained by detecting chemiluminescence with a luminometer plate reader.

## Shake-flask fermentation

Shake-batch fermentations were conducted in 50.0 mL tube with the working volume of 30.0 mL. Firstly, *S. putrefaciens* and *R. palustris* were cultured in O$_2$-free LB medium respectively. Then, 1% (v/v) seed culture of *S. putrefaciens* and *R. palustris* solution with the culture OD$_{600}$ around 0.8 were transferred respectively into the tubes containing 30 mL LB medium with or without 2 mM Fe$^{3+}$. The fermentation temperature was set at 30 °C and the agitation speed was controlled at 220 rpm. In addition, a full-spectrum LED (100 W) was used for external light illumination.

## Fed-batch fermentation

Fed-batch fermentations were conducted in 5.0 L fermenter with the working volume of 3.0 L. Firstly, *S. putrefaciens* and *R. palustris* were cultured in O$_2$-free LB medium respectively. Then, both the *S. putrefaciens* and *R. palustris* solution with the culture OD$_{600}$ around 0.8 was transferred into sterilized culture bottle with 400 mL O$_2$-free LB medium containing 2 mM Fe$^{3+}$. The *S. putrefaciens-R. palustris* co-culture seed was then incubated at 30 °C with a shaking rate of 220 rpm for 96 h, and a full-spectrum LED (100 W) was used as its light source. Subsequently, 1% (v/v) seed culture was transferred into sterilized fermenter, and the fed-batch fermentation started with feeding. An anaerobic condition inside fermenter was created by flushing argon gas until concentration of dissolved oxygen dropped to zero. In order to maintain anaerobic conditions inside fermenter, the feed medium (LB medium containing 2 mM Fe$^{3+}$) was also flushed with argon gas before being fed into fermenter. 3 g/L HCl and 3 g/L NaOH were used to control the pH of bacterial fermentation broth at -7.0. The fermentation temperature was set at 30 °C and the agitation speed was controlled at 220 rpm. In addition, a full-spectrum LED (100 W) was used for external light illumination.

## 16S rRNA gene sequencing

Total genomic DNA was extracted from the *S. putrefaciens-R. palustris* co-culture and *S. putrefaciens-R. palustris-G. Soli* microbial consortia. Three independent biological replicates were performed. Specifically, V3 and V4 regions of 16S rRNA were amplified by PCR with primers of 341 F:5′-CCTA CGGG NGGC WGCA G-3′ and 805 R: 5′-GACT ACHV GGGT ATCT AATC C-3′. The amplified DNA was then processed for gene sequencing with Illumina MiSeq-PE250 system. QIIME2 was adopted to analyze the gene sequences and classify the operational taxonomic units (OTUs) on the basis of 97% sequence identity. Subsequently, the functional potential of the bacterial community was predicted by using PICRUSt2 based on the 16 S rRNA gene sequencing data. The predictive genes were then used to construct Kyoto Encyclopedia of Genes and Genomes (KEGG) pathways based on KEGG database. Differential functional gene expression was analyzed with STAMP 2.0.

## RNA extraction, RNA-Seq and transcriptomic analysis

Three independent biological replicates of *S. putrefaciens-R. palustris* co-culture with/without Fe redox communication were subjected to RNA-seq analysis. According to the manufacturer's protocols, the total RNA was extracted using the Trizol reagent (Invitrogen, California, USA). RNA quantity was measured with the Qubit 2.0 fluorometer (ThermoFisher Scientific, Waltham, MA, USA). Obtained RNA was quality and integrity checked using the RNA 6000 Nano kit with an Agilent Bioanalyzer 2100 (Agilent Technologies). Then the rRNA was depleted with Ribo-Zero rRNA Removal Kit (bacteria) (Illumina, San Diego, CA, USA) and cDNA libraries were constructed with an Illumina NEBNext Ultra RNA Library Prep Kit (New England Biolabs, Ipswich, MA, USA). Subsequently, the cDNA libraries were pair-end-sequenced on an Illumina Novaseq 6000 platform (Illumina, San Diego, CA, USA). Afterward, Cutadapt software was adopted to clean the raw reads by removing adaptor sequences, ambiguous based annotated as N, and low-quality sequences (Q$_{phred}$ ≤ 20 bases, accounting for >50% of the total read length). The clean reads were obtained with the FastQC software and were then mapped to prjna881029 (*S. putrefaciens*) and prjna683609 (*R. palustris*) reference genome using STAR software. The read counts and fragments per kilobase of transcripts per million mapped reads were calculated using featureCounts (version 1.5.0-p3) and kallisto (version 0.46.1) respectively. Mapped reads were normalized with fragments per kilobase per million mapped reads (FPKM). Differential expression analysis of two groups was performed using DESeq2 (version 1.16.1), which is based on a negative binomial distribution model. The standardized method was DESeq. Genes with adjusted *P* value < 0.05 and Log$_2$ fold change ≥ |0.58| were considered significantly differentially expressed.

## Bioelectrochemical detection

The gold interdigitated microelectrode was adopted to measure electron transfer function of *S. putrefaciens-R. palustris* co-culture with or without Fe redox communication. The gold interdigitated microelectrode was fabricated on a silicon wafer with 300 nm thermal silicon oxide ($SiO_2/Si$) by photolithography and metal evaporation[47]. To be specific, a 4 inch silicon wafer was washed sequentially in acetone, ethanol and deionized (DI) water. Subsequently, the silicon wafer was dried with a compressed nitrogen gun. The residual moisture and solvent were removed by baking the wafer at 180 °C for 5 min. After cooling down, 3 ml of S1813 resist was dispensed at the wafer center and the wafer was then baked on the hot plate at 115 °C for 1 min. Then, a direct-write optical lithography system (ABM. Inc, America) was adopted for the preparation of the gold interdigitated microelectrode. After the photolithography process, the wafer was then loaded face down onto the sample holder of a thermal evaporator system (Jiashuo JSD-300, China) to deposit the active material (The thickness of chromium and gold are set as 5 nm and 40 nm respectively). *S. putrefaciens-R. palustris* co-culture with or without Fe redox communication incubated at 96 h were washed twice with 1×PBS buffer. Then, we fixed the $OD_{600}$ of *S. putrefaciens-R. palustris* co-culture to 3.0 with 1 × PBS buffer. Subsequently, the real-time current intensities of the co-culture were measured by injecting 10 μL bacterial suspension into the gold interdigitated microelectrode with the input voltage set at 0.2 V. Reported current densities (μA cm$^{-2}$) were then obtained by averaging the ratio of the stable current intensity to the effective working-area (0.80 × 0.88 mm) of the gold interdigitated microelectrode.

## Determination of $CO_2$ fixation rate

The $CO_2$ fixation rates in *S. putrefaciens-R. palustris* co-culture and *S. putrefaciens-R. palustris-G. Soli* microbial consortia were measured respectively. The bacteria were cultured in LB medium with the addition of $NaHCO_3$ (30 mM). Then, the suspension was collected by centrifugation at 5000 × *g* for 5 min. $H_2SO_4$ solution (98 wt%) was used to prepare a 5 mM solution for the quantification of inorganic carbon by recording the mass variation. The $CO_2$ fixation rate was calculated as: bicarbonate consumption (mM)/time interval (96 h)/DCW(g).

## Statistics and reproducibility

All data are presented as mean ± standard deviation (SD) and statistical analyses and graphs were performed using OriginLab 9.0 and GraphPad Prism 8.0.1. For comparison of two groups, significance was determined by the two-tailed unpaired Student's *t* test. Each *n* indicates the number of biologically independent samples. Data in Figs. 2c, f, h, 3c–e, 5a–d and 6d–g and Supplementary Figs. 2, 9b, e, f, 10a, b, 11, 22 and 26 were successfully replicated in two independent experiments.

## Reporting summary

Further information on research design is available in the Nature Portfolio Reporting Summary linked to this article.

## Data availability

The source data are provided as a Source Data file. The 16 S rRNA sequencing and transcriptomic datasets have been deposited in the Genome Sequence Archive (GSA) under accession numbers CRA012892 and CRA012890 respectively. Source data are provided as a Source Data file. Source data are provided with this paper.

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

## Acknowledgements

The National Natural Science Foundation of China (21925401 to Q.Y., 22322408 to Y.Y.), National Key R&D Program of China (2017YFA0208000 to Q.Y., 2021YFA1202400 to Y.Y.), the Fundamental Research Funds for the Central Universities (2042022rc0004 to Y.Y.), the New Cornerstone Science Foundation through the XPLORER PRIZE and the interdisciplinary innovative talents foundation from Renmin Hospital of Wuhan University are acknowledged for research funding. We thank the Core Facility of Wuhan University for SEM and TEM analysis.

## Author contributions

Q.Y., L.Y., and N.C. designed the project. N.C. and N.D. performed all the experiments and were primarily responsible for data collection. N.D., R.S., T. H., and J.X. drew all the schematic illustrations. Q.Y., L.Y., W.T., T. L., Y.Y., G. B., J.T., N.D., and N.C. analyzed the results and prepared the paper, Figures and Supplementary Information. All authors contributed to the discussion and editing of the paper.

## Competing interests

The authors declare no competing interests.
