## [Peer review file · Nature Communications]

REVIEWER COMMENTS

Reviewer #1 (Remarks to the Author):

The manuscript by Chen N. et al. reported an electron transfer triggered redox communication network based on Fe³⁺/Fe²⁺ redox signal interception, relay and interpretation. Specifically, first, the authors designed and verified Fe³⁺-reducing bacteria *S. putrefaciens* as router to release electron, followed by the design of *R. palustris* to ingest the electron and reconvert Fe²⁺ to Fe³⁺. Thus, bacteria *S. putrefaciens* plays as CPU, and *R. palustris* plays as verifier and actuator within the network based on Fe³⁺/Fe²⁺ redox. Overall, this is a good piece of work with smart design. Using the developed system, the authors verified the relationship between electron transfer and improved lycopene production, electron transfer and CO₂ fixation. Moreover, it is also applicable both in co-culture and microbial consortia. Finally, this work provides fundamental insights into the redox communication between microbial species and revealed the metabolism coordination mechanism, thereby showing great application potential in synthetic biology.

Other issues:

1. Line 36, authors stated that their strategy has promising in understanding life evolution, but the manuscript lacks corresponding description.
2. In the transcriptomic analysis, authors should add description about genes associated with lycopene biosynthesis or CO₂ fixation in the text or supplementary information.
3. There are some writing mistakes, authors need to check the manuscript carefully:
 - (1) The full name of the bacterium should be given when appear for the first time.
 - (2) Line 119, mutant should be mutants.
 - (3) Line123, system should be systems
 - (4) Some formatting issues in the references

Reviewer #2 (Remarks to the Author):

The authors have provided a very interesting and creative methodology for the use of redox as a means to interrogate and control a variety of microbial systems. They have introduced a "verifier" that consists of a nanoparticle methodology based on luminescence that provides the relative ratio of Fe³⁺ to Fe²⁺ and importantly, in all the examples provided, this ratio is critical to the function of the co-culture or consortia of assembled cells. I am very excited about this work and would like to see it appear in the journal. I have, however, enumerated a fairly comprehensive list of comments that might be considered. The intent of these comments is to provide a bit more justification and mainly to provide clarity in the presentation.

A list of some points

1. Abstract "...and etc" is inappropriate.
2. Abstract "...universal.." Is it a meaningful goal to have a "universal" method for microbial communication? Perhaps we should just have a variety of means to interrogate this communication and redox is one such means among many.
3. Abstract "...understanding life evolution" is lofty and not worded well. This reviewer notes that there is no data relative to evolutionary processes, so perhaps this could be struck from the text.
4. Introduction. Line 50-51. The data does not show "understanding the life evolution". While a nice goal, perhaps this could be omitted or otherwise altered to "...driving microbial industries and better

understand living systems”.

5. Intro. Line 68. “ The Fe³⁺/Fe²⁺ redox couple, which widely exists in nature, is critical in maintaining metabolic processes”. [suggested wording]

6. Intro. Line 86, the text reads that the Fe²⁺ can be transferred to *R. paulstris* through cytochrome c, pilin, and etc. Please be specific. Is the Fe²⁺ actually imported? If so, probably not by pilin. Or does it interact with the cytochrome, please be specific. I find this is a general theme throughout. There are many instances where results are given and a one sentence explanation is provided that has minimal detail. Perhaps go through the document text to ensure mechanisms are provided and sentences are clear.

7. Intro. Line 88. First use of the acronym, LAN, appears here and should be described and attributed to the original citation. It is not introduced in this manuscript. Perhaps also it would be appropriate to attribute the structure that this work is based on (LAN, CPU, etc.) to the original citation and state something like, “we have captured the concept of a LAN, CPU, and modular assembly in ref x, but applied the same methodology to a new iron based system that has these benefits...”

8. Intro. Line 90. The text says the work is done both in co-culture and in consortia. Please elucidate what the consortia is right here.

9. Figure 2, indicate approximate Fe²⁺ concentration of the red color in “b” and “e”. This can be alongside the color bar in a vertical manner.

10. Results. Lines 116 and 123. Please introduce MtrC and CymA and why they are important. See item 6 above.

11. Results. Line 124. Why is HA important and what does it do, specifically. Is there evidence for this and if so, cite this or demonstrate in Supplemental.

12. Results. Line 132. PioABC complex is reported to directly uptake Fe²⁺, please provide reference. Include references that it is converted back to Fe³⁺. Show in Figure S2 that the Fe³⁺ is generated and where. Again, refer to item 6 above.

13. Results. Figure SF3. Include what the co-culture is comprised of. This should be fraction of each cell in the inoculum as well as any time dependent data that is available. OD for each culture over time would be good here.

14. Results. Line 144. Figure 2f. Explain more carefully plus and minus Fe redox communication and “bare co-culture” system. It is not clear to me. Is iron added in one case and not in the other? This is very important as it is a central theme throughout.

15. Results. Line 146. This statement, that NADPH is higher than and provided by the “redox communication process”. Not sure there is a clear reason why the “communication process” raises NADPH, or rather that NADPH just happens to be higher when Fe is added. Also, please refer to or embellish Figure S2 with NADPH or other redox inputs. Why is it here in Line 146 that the yield is increased?

16. Results. Line 154. Explain how addition of Fe²⁺ lowers the luminescence of the ZGO:Mn (Figure S6a). Use references. Then, describe in detail these experiments. The “typical phenanthroline” method is not known to many people. Cite. This reporting structure is key to the entire paper and needs to be described in much greater detail.

17. Results. Supplemental Figure S6a. Relative CFU counts. Not sure what this is. Presumably the

experiment is performed so that the luminescence is unaffected by the presence of the cells? The absolute counts are needed and the intensity should be given relative to the actual number of bacteria. Otherwise, explain more clearly. This is a follow on to item 16 above. How fast are the luminescence measurements responding? What concentration ranges are these results to represent? Is this all linear with concentration? This figure needs substantially more data. While GSSH and GSH additions make sense to me, they are not described at all. The verifier is key and is the most innovative aspect of the work. Perhaps an entire figure should be dedicated to it, its dynamics, its output w various components in the solution, etc.

18. Results. Supplementary Figure 7. Is this supposed to track with Fe^{3+} ? Why not state this. Explain how model parameters were obtained in supplemental. Equation 1 seems not to be a differential equation. Provide first principles form and then integrated form. Provide units for all state variables so that mathematical model can be appropriately evaluated. Currently, it is more of a distraction but could provide insight. Its insight provided currently is not that clear. It might be nice to provide the model simulations that correlate with the dynamic measurements of the verifier.

19. Results. Supplementary Figure 8. This refers to color but the SEMs are not color. How are morphologies related to the results? Is this important? Perhaps these should be run with and without supplemental iron of different charge?

20. Supplemental Figure 9. Suggest green and red to go with the staining above. What does it mean Fe signal? Be specific. Include concentrations.

21. Results. Figure 3f. While relative abundance conveys the data, it is difficult to see how the total changes over time. Moreover, it is difficult to parse out how much of each strain there is. Why not just 4 bars at each time?

22. Results. Figure S10. How are these different? Are there significant differences indicated by error bars and are these truly all that much different? This is not clear to me.

23. Results. Gene expression in Figure 4. Line 240. The data describe differentially expressed genes, but do not say what the reference point is. Does this mean differentially expressed in the presence of iron versus the absence of iron? Also, I believe the authors meant to write "underrepresent[ed]". While the upregulation of these genes suggest the corresponding proteins are upregulated and, as the authors have suggested, perhaps there is a concomitant upregulation of the electron transfer function, there is no confirmatory data showing the upregulation of electron transfer. Perhaps an independent experiment here would be worthwhile.

24. Results. Line 260. Here is the first time that the control or reference state was provided. That is that the differential response is compared to the "bare" co-culture. This should be stated above as well. In general, however, these transcript data offer some insight, but no solid conclusions about function. Therefore, these should be relegated to the Supplemental sections. Broad based comments on differential gene expression used to be ok in journals like J. Bact, but now they need to be independently validated and linked to specific and independent functional measurements.

25. Results. Lines 288-292. The data for rotenone and CCCP should be provided in reference to where they block electron transport, much like the figure provided in Supplemental Figure 13. I'm not sure what supplemental figure 13 does, if there are no experiments with knockout mutants of one or two of these pathway components (ie., P_{io}B, P_{io}A, etc.). Conversely, how are acetate and lactate electron donors mechanistically expected to improve pathway flux? This is not provided. Do they donate electrons to the Fe^{3+} ? Or are they contributors to respiratory components?

26. I like the data from Figure 5c-d. These show an altered metabolic state when cultivated with the iron. It would be good, however, as noted above to recapitulate this on a figure, such as Figure 5e,

where the role of the ratio is not entirely mechanistically clear. How specifically is NADPH/NAD to alter those specific genes?

27. The results on CO₂ sequestration and consortia function are awesome. Congratulations. All of the results are dynamite and I am thrilled to provide these comments. This reviewer would appreciate a more clear message, however, that it is the Fe²⁺ to Fe³⁺ conversion that occurs at the verifier is a key driver and also indicator. For example, in Figure 6b, the intensity of the verifier is provided, but I don't see the controls (ie., as shown in Supplemental Figure 6a)? Perhaps it would be good to have a figure in Supplemental that includes media from these experiments that is added to the Mn, the Mn³⁺, and the Mn²⁺ solutions from the SF6a? This could indicate that the verifier is working externally and the trends expected would be shown?

Reviewer #3 (Remarks to the Author):

Please see below.

The paper is capable to drive coordinated functions through sensing, analyzing and processing signal information, playing critical roles in biomanufacturing, life evolution and etc. Although it is likely to be of interest and use to readers and is not for publication with this version. I strongly suggested that it is rejection.

The following comments should be revised as the follows:

1. In the section of introduction and discussion, the authors are suggested to compare the difference between the previous studies which relate to redox-based molecular communication (such as DOI: 10.1038/ncomms14030; DOI: 10.1038/s41565-021-00878-4) with this research. What is the novelty of this research compared to previous related studies? The description of lycopene should be added in the section of introduction.
2. Is this redox-based molecular communication appropriate for fed-batch fermentation at laboratory scale to produce lycopene? How stable is this redox communication?
3. What impact do the various inoculation ratios of the strains *R. palustris* and *S. putrefaciens* have on this redox-based molecular communication and the production of lycopene? Additionally, when cultivation time increases, *R. palustris* proportion progressively rises while *S. putrefaciens* proportion gradually falls. Will this alter *R. palustris*'s receive to electrons?
4. The fermentation procedure of lycopene should be detailly described in the section of Method.
5. By regulating the expression strength of transhydrogenase, pentose phosphate pathway related genes, etc., numerous studies have successfully increased the intracellular NAD(P)H level. Although the redox-based communication can improve

the reducing power by transferring electrons, it is too complex to be used in industrial production, and it does not seem to be able to precisely regulate intracellular genes expression?

Point-to-Point Responses and List of Revisions

We thank reviewers for the positive endorsement and the highly insightful comments about our study. We have carefully considered the review comments and thoroughly revised our manuscript to fully address these comments, as detailed in the point-to-point response below.

Response to Reviewer #1:

Reviewer comments: The manuscript by Chen N. et al. reported an electron transfer triggered redox communication network based on Fe³⁺/Fe²⁺ redox signal interception, relay and interpretation. Specifically, first, the authors designed and verified Fe³⁺-reducing bacteria *S. putrefaciens* as router to release electron, followed by the design of *R. palustris* to ingest the electron and reconvert Fe²⁺ to Fe³⁺. Thus, bacteria *S. putrefaciens* plays as CPU, and *R. palustris* plays as verifier and actuator within the network based on Fe³⁺/Fe²⁺ redox. Overall, this is a good piece of work with smart design. Using the developed system, the authors verified the relationship between electron transfer and improved lycopene production, electron transfer and CO₂ fixation. Moreover, it is also applicable both in co-culture and microbial consortia. Finally, this work provides fundamental insights into the redox communication between microbial species and revealed the metabolism coordination mechanism, thereby showing great application potential in synthetic biology.

Response: We sincerely thank the reviewer for carefully reading our manuscript with highly constructive comments, and supporting the publication in *Nature Communications*. We greatly appreciate the insightful comments and suggestions, and have made the corresponding changes accordingly.

Specific Comment 1. Line 36, authors stated that their strategy has promising in understanding life evolution, but the manuscript lacks corresponding description.

Response and Revisions: We greatly appreciate the reviewer's valuable suggestions. As the reviewer has correctly mentioned, the manuscript lacks corresponding description in understanding life evolution and the description in the original manuscript might be ambiguous. To make the descriptions more precise, we have deleted the misleading words "understanding life evolution" and revised the description to "promising in offering insight for improving biomanufacturing efficiency" in the revised manuscript.

Specific Comment 2. In the transcriptomic analysis, authors should add description about genes associated with lycopene biosynthesis or CO₂ fixation in the text or supplementary information.

Response and Revisions: Thanks for the reviewer's valuable suggestion. According to reviewer's suggestion, we added the description about genes associated with lycopene biosynthesis or CO₂ fixation "Besides, Fig. 4a also shows that the Log₂ FC

of proteins associated with both the MEP & MVA pathways and CO₂ fixation are above 0 generally (Supplementary Fig. 16), suggesting that the redox communication triggered with electron transfer facilitates the biosynthesis and CO₂ fixation processes in microorganism. ” in the revised manuscript.

Specific Comment 3. There are some writing mistakes, authors need to check the manuscript carefully:

- (1) The full name of the bacterium should be given when appear for the first time.
- (2) Line 119, mutant should be mutants.
- (3) Line123, system should be systems
- (4) Some formatting issues in the references

Response and Revisions: Thanks for the reviewer’s valuable suggestions. We have carefully read through the whole manuscript and revised the spelling and formatting mistakes highlighted in yellow. According to reviewer’s suggestion, the full name of the bacterium were given when appear for the first time (Page 4, 1st paragraph in introduction) . Furthermore, we corrected the descriptions of “mutant” with “ mutants” and “system” with “ systems”. Formatting issues in the references were also corrected highlighted in the revised manuscript. _____

Response to Reviewer #2:

Reviewer Comments: The authors have provided a very interesting and creative methodology for the use of redox as a means to interrogate and control a variety of microbial systems. They have introduced a “verifier” that consists of a nanoparticle methodology based on luminescence that provides the relative ratio of Fe³⁺ to Fe²⁺ and importantly, in all the examples provided, this ratio is critical to the function of the co-culture or consortia of assembled cells. I am very excited about this work and would like to see it appear in the journal. I have, however, enumerated a fairly comprehensive list of comments that might be considered. The intent of these comments is to provide a bit more justification and mainly to provide clarity in the presentation.

Response: We sincerely thank the reviewer for carefully reading our manuscript with highly constructive comments, and supporting the publication in *Nature Communications*. Furthermore, we deeply appreciate the insightful questions/suggestions, which have motivated us to conduct additional studies to further strengthen our manuscript.

Specific Comment 1. Abstract “...and etc” is inappropriate.

Response and Revisions: We greatly appreciate the reviewer’s valuable comment. Inspired by the reviewer’s comment, we deleted “and etc” and revised the description to “biomanufacturing and life evolution” (Page 1, 1st paragraph in abstract).

Specific Comment 2. Abstract “...universal..” Is it a meaningful goal to have a “universal” method for microbial communication? Perhaps we should just have a variety of means to interrogate this communication and redox is one such means among many.

Response and Revisions: We greatly appreciate the reviewer’s comments. Inspired by the reviewer’s comments, we modified the description “simple and universal” with “effective” (Page 1, 1st paragraph in abstract) in the revised manuscript.

Specific Comment 3. Abstract “...understanding life evolution” is lofty and not worded well. This reviewer notes that there is no data relative to evolutionary processes, so perhaps this could be struck from the text.

Response and Revisions: We greatly appreciate the reviewer’s valuable suggestions. According to the reviewer’s suggestion, we have deleted the misleading words “understanding life evolution” in the revised manuscript.

Specific Comment 4. Introduction. Line 50-51. The data does not show “understanding the life evolution”. While a nice goal, perhaps this could be omitted or otherwise altered to “...driving microbial industries and better understand living systems”.

Response and Revisions: Thanks for the reviewer’s valuable suggestions. According to the reviewer’s suggestion, we altered the sentences to “...driving microbial industries and better understand living systems” in the revised manuscript (Page 3, 1st paragraph in introduction).

Specific Comment 5. Intro. Line 68. “ The Fe³⁺/Fe²⁺ redox couple, which widely exists in nature, is critical in maintaining metabolic processes”. [suggested wording]

Response and Revisions: Thanks for the reviewer’s valuable suggestions. According to the reviewer’s suggestion, we revised the sentences “Fe³⁺/Fe²⁺ redox couple, widely existed in nature, is critical in mediating metabolism process” to “The Fe³⁺/Fe²⁺ redox couple, which widely exists in nature, is critical in maintaining metabolic processes” in the revised manuscript (Page 3, 3rd paragraph in introduction).

Specific Comment 6. Intro. Line 86, the text reads that the Fe²⁺ can be transferred to *R. paulstris* through cytochrome c, pilin, and etc. Please be specific. Is the Fe²⁺ actually imported? If so, probably not by pilin. Or does it interact with the cytochrome, please be specific. I find this is a general theme throughout. There are many instances where results are given and a one sentence explanation is provided that has minimal detail. Perhaps go through the document text to ensure mechanisms are provided and sentences are clear.

Response and Revisions: We appreciate the reviewer’s valuable suggestion. Inspired by the reviewer’s suggestion, we modified descriptions to “Actually, electrons from Fe²⁺ redox signal can be transferred into *R. palustris* bio-actuator cells with iron oxidation pathway through the electron transfer related proteins (Supplementary Fig. 4).” (Page 4, 1st paragraph in introduction). Scheme of the iron oxidation pathway was provided here in Supplementary Fig. 4 and related reference was added.

Furthermore, we added description “Specifically, PioA and PioB are proposed to oxidize Fe²⁺ extracellularly and then transfer the released electrons across the outer membrane to PioC, which is hypothesized to be in the periplasm.” (Page 17, 1st paragraph in results) to describe clearly how the electrons of Fe²⁺ transfer to *R. paulstris* through iron oxidation pathway.

Specific Comment 7. Intro. Line 88. First use of the acronym, LAN, appears here and should be described and attributed to the original citation. It is not introduced in this manuscript. Perhaps also it would be appropriate to attribute the structure that this work is based on (LAN, CPU, etc.) to the original citation and state something like, “we have captured the concept of a LAN, CPU, and modular assembly in ref x, but applied the same methodology to a new iron based system that has these benefits...”

Response and Revisions: Thanks for the reviewer’s valuable suggestions. According to the reviewer’s suggestions, we added related descriptions “We then capture the concept of biological local area network (LAN) which is consisted of “router” “actuator” and “verifier” as demonstrated in reference¹, and applied the same methodology to a new Fe signal based system that has these benefits” in the revised manuscript (Page 4, 1st paragraph in introduction).

Specific Comment 8. Intro. Line 90. The text says the work is done both in co-culture and in consortia. Please elucidate what the consortia is right here.

Response and Revisions: We appreciate the reviewer's suggestion. According to the reviewer's suggestion, we added descriptions "*S. putrefaciens-Geobacter soli-R. palustris*" to elucidate what the consortia is (Page 5, 1st paragraph in introduction).

Specific Comment 9. Figure 2, indicate approximate Fe²⁺ concentration of the red color in "b" and "e". This can be alongside the color bar in a vertical manner.

Response and Revisions: Thanks for the reviewer's valuable suggestions. According to the reviewer's suggestion, we added " $C_{\text{Fe}^{2+}}$ (mM)" alongside the color bar in a vertical manner to indicate Fe²⁺ concentration in Fig. 2b and 2e in the revised manuscript.

Specific Comment 10. Results. Lines 116 and 123. Please introduce MtrC and CymA and why they are important. See item 6 above.

Response and Revisions: We appreciate the reviewer's suggestion. According to the reviewer's suggestion, we added descriptions "To be specific, Mtrc, OmcA and OmcB locate on the bacteria surface and could transfer electrons directly²², and CymA is a key membrane-anchor protein that could transfer electrons from cytoplasm to periplasm in *S. putrefaciens*²². " (Page 6, 2nd paragraph in results) to introduce MtrC and CymA. Furthermore, related references were also added for better understanding.

Specific Comment 11. Results. Line 124. Why is HA important and what does it do, specifically. Is there evidence for this and if so, cite this or demonstrate in Supplemental.

Response and Revisions: Thanks for the reviewer's valuable suggestions. According to the reviewer's suggestion, we cited here to provide evidence for this sentence (Page 7, 1st paragraph in results).

Specific Comment 12. Results. Line 132. PioABC complex is reported to directly uptake Fe²⁺, please provide reference. Include references that it is converted back to Fe³⁺. Show in Figure S2 that the Fe³⁺ is generated and where. Again, refer to item 6 above.

Response and Revisions: We greatly appreciate the reviewer's valuable suggestions. Inspired by the reviewer's suggestions, we provided additional reference to illustrate that PioABC complex is reported to directly uptake electrons from Fe²⁺ (Page 7, 2nd paragraph in results). Furthermore, we added description "Specifically, PioA and PioB are proposed to oxidize Fe²⁺ extracellularly and then transfer the released electrons across the outer membrane to PioC, which is hypothesized to be in the periplasm." (Page 17, 1st paragraph in results) to describe clearly how the electrons of Fe²⁺ transfer to *R. palustris* through iron oxidation pathway. Besides, we modified Figure S4 (original Figure S2), showing where the reducing power NADPH is needed.

Specific Comment 13. Results. Figure S3. Include what the co-culture is comprised of. This should be fraction of each cell in the inoculum as well as any time dependent data that is available. OD for each culture over time would be good here.

Response and Revisions: Thanks for the reviewer’s valuable suggestions. Inspired by the reviewer’s suggestions, we replaced the description “co-culture” with “co-culture (*S. putrefaciens*- *R. palustris*)” in legend of Figure S5a (original Figure S3). Furthermore, according to the reviewer’s suggestion, OD curve for each culture over time was added in Figure S5b (original Figure S3) in the revised manuscript.

Specific Comment 14. Results. Line 144. Figure 2f. Explain more carefully plus and minus Fe redox communication and “bare co-culture” system. It is not clear to me. Is iron added in one case and not in the other? This is very important as it is a central theme throughout.

Response and Revisions: Thanks for the reviewer’s valuable suggestions. Inspired by the reviewer’s suggestion, we modified description “with or without Fe redox communication represented by “+” and “-” respectively” in the legend of Fig. 2f to explain plus and minus Fe redox communication. Furthermore, “bare co-culture” was modified to “the bare co-culture system without Fe redox communication” (Page 8, 1st paragraph in results) to make the description clear.

Specific Comment 15. Results. Line 146. This statement, that NADPH is higher than and provided by the “redox communication process”. Not sure there is a clear reason why the “communication process” raises NADPH, or rather that NADPH just happens to be higher when Fe is added. Also, please refer to or embellish Figure S2 with NADPH or other redox inputs. Why is it here in Line 146 that the yield is increased?

Response and Revisions: We greatly appreciate the reviewer’s valuable suggestions. Inspired by the reviewer’s suggestions, we embellish Figure S4 (original Figure S2) with NADPH to illustrate the role of NADPH in the MVA and MEP pathway. Furthermore, we modified descriptions with “Considering that the reduced nicotinamide adenine dinucleotide phosphate (NADPH) is critical in biosynthetic pathways serving as the electron carrier for a large subset of oxidoreductases²⁵, the increased lycopene yield is partially due to the enhanced NADPH in *R. palustris* by receiving electrons from the dynamic Fe³⁺/Fe²⁺ cyclic communication process through the iron oxidation pathway.(Supplementary Fig. 7)¹⁹” (Page 8, 1st paragraph in results) to illustrate the possible reason why the yield is increased and why NADPH raises.

Specific Comment 16. Results. Line 154. Explain how addition of Fe²⁺ lowers the luminescence of the ZGO:Mn (Figure S6a). Use references. Then, describe in detail these experiments. The “typical phenanthroline” method is not known to many people. Cite. This reporting structure is key to the entire paper and needs to be described in much greater detail.

Response and Revisions: Thanks for the reviewer’s valuable suggestions. Inspired by the reviewer’s suggestions, we cited here and modified descriptions to “Specifically, the luminescence of Zn₂GeO₄:Mn (ZGO:Mn) at 536 nm originates from the ⁴T₁(G)→⁶A₁(S) transition of Mn²⁺ luminescence center²². Owing to the reduction Fermi energy level location of $E_{\text{Fe}^{3+},\text{red}}$ and $E_{\text{Fe}^{2+},\text{red}}$, the nano verifier shows different luminescence responses towards Fe³⁺ and Fe²⁺, thus achieving a dynamic and reversal monitoring of

the Fe redox signal conversion in real-time (Supplementary Fig. 9)²². Additionally, the nano verifier exhibits low toxicity both against *S. putrefaciens* and *R. palustris* according to the cell viability tests (Supplementary Fig. 10). ” (Page 8, 1st paragraph in results). These experiments were described in detail in supporting information. Furthermore, according to the reviewer’s suggestion, we cited references for “typical phenanthroline” method and in the revised manuscript.

Specific Comment 17. Results. Supplemental Figure S6a. Relative CFU counts. Not sure what this is. Presumably the experiment is performed so that the luminescence is unaffected by the presence of the cells? The absolute counts are needed and the intensity should be given relative to the actual number of bacteria. Otherwise, explain more clearly. This is a follow on to item 16 above. How fast are the luminescence measurements responding? What concentration ranges are these results to represent? Is this all linear with concentration? This figure needs substantially more data. While GSSH and GSH additions make sense to me, they are not described at all. The verifier is key and is the most innovative aspect of the work. Perhaps an entire figure should be dedicated to it, its dynamics, its output w various components in the solution, etc.

Response and Revisions: We greatly appreciate the reviewer’s valuable suggestions. Inspired by the reviewer’s suggestion, we added description “Additionally, the nano verifier exhibits low toxicity both against *S. putrefaciens* and *R. palustris* according to quantitative colony forming unit (CFU) survival assays (Supplementary Fig. 10)” to avoid the uncertainty of the CFU. Considering the original Figure S6b-c aims to demonstrate the biocompatibility of the nano verifier, we replaced the two figures into Supplementary Fig. 10. Furthermore, according to the reviewer’s suggestions, we performed additional studies, aiming to provide solid information on the excellent performance of the nano verifier in Supplementary Fig. 9 in the revised supporting information.

Specific Comment 18. Results. Supplementary Figure 7. Is this supposed to track with Fe³⁺? Why not state this. Explain how model parameters were obtained in supplemental. Equation 1 seems not to be a differential equation. Provide first principles form and then integrated form. Provide units for all state variables so that mathematical model can be appropriately evaluated. Currently, it is more of a distraction but could provide insight. Its insight provided currently is not that clear. It might be nice to provide the model simulations that correlate with the dynamic measurements of the verifier.

Response and Revisions: Thanks for the reviewer’s valuable suggestions. Inspired by the reviewer’s suggestion, we modified the description in supporting information to explain how model parameters were obtained. Considering that the persistent luminescence intensity data in Supplementary Figure 11 (original Supplementary Figure 7) were adopted to fit and obtain model parameters, we replaced its location in the revised manuscript. Furthermore, we modified the description to “Furthermore, based on the recorded luminescence intensity of the verifier (Supplementary Fig. 11) and the linear relationship between logarithm of persistent luminescence intensity and

$\text{Fe}^{2+}/\text{Fe}_{\text{total}}$ ratio (Supplementary Fig. 9d), the concentrations of $\text{Fe}^{3+}/\text{Fe}^{2+}$ redox couple were calculated. With the calculated values, we then simulated the kinetic transformation of the $\text{Fe}^{3+}/\text{Fe}^{2+}$ redox signal according to the equations (Supplemental methods) with MATLAB software^{1,27} to avoid misunderstanding. Besides, the relationship between the persistent luminescence of nano verifier and concentration of Fe species was provided in the revised supporting information.

Specific Comment 19. Results. Supplementary Figure 8. This refers to color but the SEMs are not color. How are morphologies related to the results? Is this important? Perhaps these should be run with and without supplemental iron of different charge?

Response and Revisions: Thanks for the reviewer's valuable comments. Here, SEMs and Fig. 3a both aim to provide visual data for *S. putrefaciens* and *R. palustris*. Although the SEMs are capable to show the rod-shaped morphologies of bacteria, it is quite confusing owing to simultaneously referring to both color and morphologies. Inspired by the reviewer's comments, we relocate the SEMs in the revised supporting information as Supplementary Figure 1 and Supplementary Figure 3.

Specific Comment 20. Supplemental Figure 9. Suggest green and red to go with the staining above. What does it mean Fe signal? Be specific. Include concentrations.

Response and Revisions: Thanks for the reviewer's valuable suggestions. Inspired by the reviewer's suggestion, we modified the legend descriptions to "OD₆₀₀ of the *S. putrefaciens*-*R. palustris* co-culture with or without Fe redox communication ("with Fe signal" represent the addition of Fe^{3+} with the initial concentration of 2 mM, "with Fe signal" represent no addition of Fe^{3+})." in Supplemental Figure 12 (original Supplemental Figure 9) to make it specific and understandable.

Specific Comment 21. Results. Figure 3f. While relative abundance conveys the data, it is difficult to see how the total changes over time. Moreover, it is difficult to parse out how much of each strain there is. Why not just 4 bars at each time?

Response and Revisions: We greatly appreciate the reviewer's valuable suggestions. Inspired by the reviewer's suggestions, we added a new figure in which the OD₆₀₀ of each strain were demonstrated at each time (Supplemental Figure 9).

Specific Comment 22. Results. Figure S10. How are these different? Are there significant differences indicated by error bars and are these truly all that much different? This is not clear to me.

Response and Revisions: Thanks for the reviewer's valuable suggestions. Inspired by the reviewer's suggestion, we added description "while the OD₆₀₀ values of *S. putrefaciens* are similar with the values in bare co-culture system across time (Supplementary Fig. 14)." (Page 12, 1st paragraph in results) to illustrate clearly the Supplementary Fig. 14 (original Figure S10).

Specific Comment 23. Results. Gene expression in Figure 4. Line 240. The data describe differentially expressed genes, but do not say what the reference point is. Does this mean differentially expressed in the presence of iron versus the absence of

iron? Also, I believe the authors meant to write “underrepresent[ed]”. While the upregulation of these genes suggest the corresponding proteins are upregulated and, as the authors have suggested, perhaps there is a concomitant upregulation of the electron transfer function, there is no confirmatory data showing the upregulation of electron transfer. Perhaps an independent experiment here would be worthwhile.

Response and Revisions: We greatly appreciate the reviewer’s valuable suggestions. Inspired by the reviewer’s suggestion, we modified the description to “Differentially expressed genes were evaluated according to the Log₂ fold change (Log₂ FC), which was calculated with $\{\text{Log}_2 [(\text{Gene read counts})_{(\text{Experimental})}] - \text{Log}_2 [(\text{Gene read counts})_{(\text{Control})}]\}$.” to illustrate that Log₂ FC means differentially expressed in the presence of iron versus the absence of iron. Furthermore, we added description “with the reference of gene expressions in bare co-culture.” (Page 14, 1st paragraph in results) to stress the reference point.

Specific Comment 24. Results. Line 260. Here is the first time that the control or reference state was provided. That is that the differential response is compared to the “bare” co-culture. This should be stated above as well. In general, however, these transcript data offer some insight, but no solid conclusions about function. Therefore, these should be relegated to the Supplemental sections. Broad based comments on differential gene expression used to be ok in journals like J. Bact, but now they need to be independently validated and linked to specific and independent functional measurements.

Response and Revisions: Thanks for the reviewer’s valuable suggestions. Inspired by the reviewer’s suggestion, we performed additional functional measurements and added relative description “To validate the increased electron transfer functions, current density of cells were measured with gold interdigitated microelectrode (Supplemental methods). The results demonstrate that cells in the biological LAN with redox communication have about 2.10 times higher current density compared with cells without redox communication, suggesting the improved electron transfer capability (Supplementary Fig. 15).”(Page 14, 1st paragraph in results).

Specific Comment 25. Results. Lines 288-292. The data for rotenone and CCCP should be provided in reference to where they block electron transport, much like the figure provided in Supplemental Figure 13. I’m not sure what supplemental figure 13 does, if there are no experiments with knockout mutants of one or two of these pathway components (ie., PioB, PioA, etc.). Conversely, how are acetate and lactate electron donors mechanistically expected to improve pathway flux? This is not provided. Do they donate electrons to the Fe³⁺? Or are they contributors to respiratory components?

Response and Revisions: We greatly appreciate the reviewer’s valuable suggestions. According to the reviewer’s suggestion, we provided two additional figures (Supplemental Figure 20 and 21), aiming to illustrate where they block electron transport. Furthermore, we modified description to “through the addition of electron

donors for microbial respiration including acetate and lactate” (Page 18, 1st paragraph in results) to avoid misunderstanding.

Specific Comment 26. I like the data from Figure 5c-d. These show an altered metabolic state when cultivated with the iron. It would be good, however, as noted above to recapitulate this on a figure, such as Figure 5e, where the role of the ratio is not entirely mechanistically clear. How specifically is NADPH/NAD to alter those specific genes?

Response and Revisions: Thanks for the reviewer’s valuable suggestions. Actually, we mean to illustrate that the supply of the electrons to *R. palustris* provides the critical reducing power NADPH to drive the biosynthesis process. The original descriptions might have been misunderstanding. Inspired by the reviewer’s suggestion, we modified the description to “ It is widely recognized that NADPH acts as critical electron carrier for a large subset of oxidoreductases to drive the microbial biosynthesis metabolism²⁵. Hence, the lycopene biosynthesis yields can be improved through the MVA and MEP pathway in *R. palustris* with the continuous supply of electrons.” (Page 18, 2nd paragraph in results) for better understanding.

Specific Comment 27. The results on CO₂ sequestration and consortia function are awesome. Congratulations. All of the results are dynamite and I am thrilled to provide these comments. This reviewer would appreciate a more clear message, however, that it is the Fe²⁺ to Fe³⁺ conversion that occurs at the verifier is a key driver and also indicator. For example, in Figure 6b, the intensity of the verifier is provided, but I don’t see the controls (ie., as shown in Supplemental Figure 6a)? Perhaps it would be good to have a figure in Supplemental that includes media from these experiments that is added to the Mn, the Mn³⁺, and the Mn²⁺ solutions from the SF6a? This could indicate that the verifier is working externally and the trends expected would be shown?

Response and Revisions: Thanks for the reviewer’s valuable suggestions. According to the reviewer’s suggestion, we added control groups in Figure 6b. Furthermore, we added new figures in Supplemental Figure 9b (original Supplemental Figure 6a) that includes media that is added to the Mn, the Mn⁴⁺, and the Mn²⁺ solutions to indicate that the verifier is working externally.

Response to Reviewer #3:

Reviewer Comments: The paper is capable to drive coordinated functions through sensing, analyzing and processing signal information, playing critical roles in biomanufacturing, life evolution and etc. Although it is likely to be of interest and use to readers and is not for publication with this version. I strongly suggested that it is rejection. The following comments should be revised as the follows:

Response: We sincerely thank the reviewer for carefully reading our manuscript with highly constructive comments. Furthermore, we deeply appreciate the insightful questions/suggestions, which have motivated us to conduct additional studies to further strengthen our manuscript for publication .

Specific Comment 1. In the section of introduction and discussion, the authors are suggested to compare the difference between the previous studies which relate to redox-based molecular communication (such as DOI: 10.1038/ncomms14030; DOI: 10.1038/s41565-021- 00878-4) with this research. What is the novelty of this research compared to previous related studies? The description of lycopene should be added in the section of introduction.

Response and Revisions: Thanks for the reviewer’s valuable suggestions. Inspired by the reviewer’s suggestion, we added description “Compared with the previous work on redox-based communication^{1,13}, the electron transfer triggered redox communication strategy proposed in this study was independent of external electronics input and the intrinsic electron transfer triggered redox cycle could function as batteries that support microbial metabolism²⁰.” (Page 5, 1st paragraph in results), “From another perspective, this work proposed a new redox-based communication network strategy that was independent of external electronics input and the intrinsic electron transfer triggered redox cycle could function as batteries that facilitate the biosynthesis processes. ” (Page 23, 1st paragraph in discussion) and “we believe that the proposed redox communication strategy may pave new ways for self-adaption and dynamic microbial metabolism regulation and is promising in facilitating the development of mixed fermentation industries with the distinct advantages of alleviating metabolic burden by division of labor.” (Page 23, 1st paragraph in discussion) to point out the novelty in this research.

Specific Comment 2. Is this redox-based molecular communication appropriate for fed-batch fermentation at laboratory scale to produce lycopene? How stable is this redox communication?

Response and Revisions: Thanks for the reviewer’s valuable suggestions. Inspired by the reviewer’s suggestions, we performed fed-batch fermentation at laboratory scale to produce lycopene (Supplementary Fig. 23). Actually, we aim to explore the possibility of our strategy in fed-batch fermentation here and the fed-batch fermentation process parameters in this work have not optimized yet. We believe that the fed-batch fermentation yield might be higher with optimized parameters. Base on the fed-batch fermentation results, we added description “Subsequently, fed-batch fermentation at laboratory scale was further performed to explore the applicability of

the proposed redox based communication strategy. It is inspiring to find that the lycopene biosynthesis increased about 1.43 times in fed-batch fermentation compared with the shake-flask fermentation (Supplementary Fig. 23). The above results indicated the potential application prospects of the proposed redox based communication strategy in practical lycopene biomanufacturing.” (Page 18, 2nd paragraph in discussion). Additionally, we added a new figure to demonstrate the stability of the redox communication in lycopene production (Supplementary Fig. 23c).

Specific Comment 3. What impact do the various inoculation ratios of the strains *R. palustris* and *S. putrefaciens* have on this redox-based molecular communication and the production of lycopene? Additionally, when cultivation time increases, *R. palustris* proportion progressively rises while *S. putrefaciens* proportion gradually falls. Will this alter *R. palustris*'s receive to electrons?

Response and Revisions: Thanks for the reviewer’s valuable suggestions. Inspired by the reviewer’s suggestion, we added a new figure to illustrate the impact of various inoculation ratios of the strains *R. palustris* and *S. putrefaciens* on the production of lycopene (Supplementary Fig.6).

As demonstrated in Supplementary Fig. 11, we can observe that the persistent luminescence decreased gradually in *S. putrefaciens-R. palustris* co-culture, suggesting the oxidation of Fe^{2+} into Fe^{3+} by *R. palustris* (Supplementary Fig. 9d). We can also observe that although the *S. putrefaciens* proportion gradually falls, the OD₆₀₀ of *S. putrefaciens* remain stable. Considering the above results, it can be conclude that the gradually decreased *S. putrefaciens* proportion will not alter *R. palustris*'s receive to electrons.

Specific Comment 4. The fermentation procedure of lycopene should be detailly described in the section of Method.

Response and Revisions: Thanks for the reviewer’s valuable suggestions. According to the reviewer’s suggestions, we added descriptions on the fermentation procedure of lycopene in the section of Method in the revised manuscript to make this manuscript more readable.

Specific Comment 5. By regulating the expression strength of transhydrogenase, pentose phosphate pathway related genes, etc., numerous studies have successfully increased the intracellular NAD(P)H level. Although the redox-based communication can improve the reducing power by transferring electrons, it is too complex to be used in industrial production, and it does not seem to be able to precisely regulate intracellular genes expression?

Response and Revisions: Thanks for the reviewer’s valuable comments. As the reviewer has correctly mentioned that by regulating the expression strength of transhydrogenase, pentose phosphate pathway related genes, etc., the intracellular NAD(P)H level can be increased. Actually, various methods have been demonstrated

to increase the intracellular NADPH level, and in this work we proposed a new strategy from another perspective in co-culture system. The original description might be unable to indicate the novelty and applicability. Inspired by the reviewer's comments, we have performed fed-batch fermentation at laboratory scale to investigate the possible applicability in industries. Inspiringly, the fed-batch fermentation results shows 1.43 times higher yield of lycopene compared with the shake-batch fermentations. Additionally, we modified description both in introduction and discussion to highlight the novelty of this work by comparing with previous work in the revised manuscript.

REVIEWERS' COMMENTS

Reviewer #1 (Remarks to the Author):

I have no further comments on this revised manuscript.

Reviewer #2 (Remarks to the Author):

The authors have addressed my concerns.

Reviewer #3 (Remarks to the Author):

I am satisfied with the revised manuscript. I agree that the paper will be published in Nature Communications.